# DOES LLM PRE-TRAINING TYPICALLY OCCUR AT THE EDGE OF STABILITY?

**Yuhang Cai**[*†]    **Haofeng Huang**[*‡]    **Haodong Wen**[*‡]    **Deyi Liu**[§]    **Yiyuan Ma**[§]    **Kaifeng Lyu**[‡]

## ABSTRACT

Quadratic approximations are a common lens for neural network optimization, but recent evidence challenges their predictive validity. In full-batch gradient descent with LR $\eta$, Cohen et al. (2021) observed the Edge of Stability (EoS), where the largest Hessian eigenvalue concentrates near $2/\eta$, in tension with classical stability conditions. In this work, we revisit the fidelity of quadratic approximation as a model of neural network training dynamics, with particular focus on its failure modes in LLM training. We first identify and decouple a distinct failure mechanism of the quadratic approximation regardless of the LR choice, which arises from persistent negative curvature during training, which we term the *Edge of Convexity* (EoC). Based on the decoupling from EoC, we then extend the definition of EoS to large-scale stochastic training with adaptive optimizers. Across different LLM pretraining with various model sizes up to 1.7B, we find: (1) EoC is always observed across LLM pretraining. (2) EoS is also prevalent but not universal; it disappears when the LR becomes sufficiently small (e.g., after decay) or when the batch size falls below a critical threshold that is linearly related to the critical batch size. Together, these findings characterize when and how quadratic approximations fail and serve as foundations for future work on understanding the training dynamics of modern neural networks.

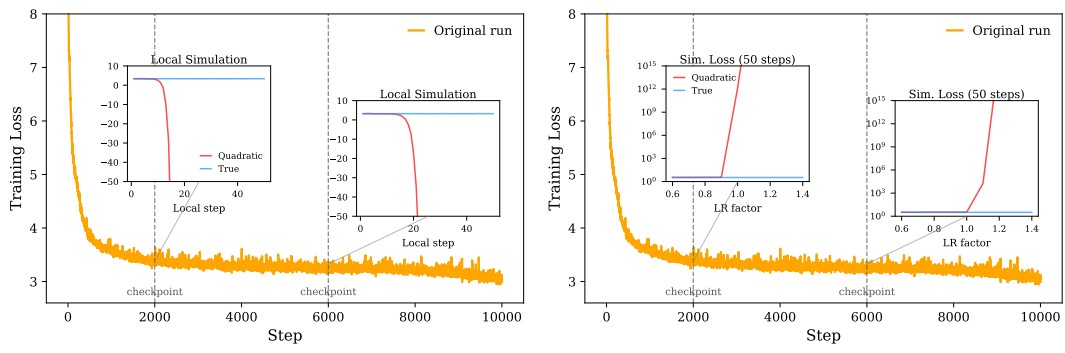

(a) Illustration of Edge of Convexity (EoC).    (b) Illustration of Edge of Stability (EoS).

Figure 1: **Edge of Convexity (EoC) and Edge of Stability (EoS).** We train a 130M GPT-like model on FinWeb-10B and run local simulations at selected checkpoints. (a) The quadratic approximation diverges while the true dynamics remain stable. (b) With convexity compensation, a small LR increase induces a sharp loss transition, revealing the quadratic instability boundary.

In optimization theory, the dynamics of gradient-based methods are commonly analyzed through a local *quadratic approximation* of the loss function, given by the second-order Taylor expansion around the current iterate. This approximation yields the well-known *descent lemma* for gradient descent, which guarantees a decrease in loss when the learning rate (LR) $\eta$ satisfies $\eta < 2/\lambda_{\max}$. Here, $\lambda_{\max}$ is the maximum eigenvalue of Hessian, which serves as a measure of the local curvature of the loss function. A standard analysis pipeline in optimization theory then proceeds by bounding quantities that appear in the lemma and deriving guarantees on the loss convergence.

However, in deep learning, the quadratic approximation often fails to accurately capture the true training dynamics. Notably, Cohen et al. (2021) show that full-batch gradient descent (GD) typically

---
[*]Equal contribution.

[†]Affiliations: † UC Berkeley; ‡ Tsinghua University; § ByteDance Seed.

[‡]Correspondence: willcai@berkeley.edu, klyu@mail.tsinghua.edu.cn.

operates at the *Edge of Stability* (EoS) when training neural networks. For a fixed learning rate $\eta$, the maximum Hessian eigenvalue $\lambda_{\max}$ tends to increase during training and eventually oscillates around $2/\eta$, thus $\eta \approx 2/\lambda_{\max}$. Under the quadratic approximation, this places the dynamics exactly at the stability boundary of gradient descent, since the quadratic model predicts loss divergence once $\eta > 2/\lambda_{\max}$. Consequently, the descent lemma can no longer guarantee a persistent decrease of the loss. Yet in practice, the loss continues to decrease over long time scales. Similar EoS behavior has also been observed for full-batch adaptive optimizers such as Adam and RMSProp (Cohen et al., 2022), when curvature is measured using an appropriately preconditioned Hessian.

The EoS phenomenon therefore exposes a fundamental limitation of existing optimization analyses for deep learning, as many such analyses critically rely on the descent lemma and the underlying quadratic approximation. This discrepancy has motivated a line of work that seeks to develop tools to understand training dynamics without critically relying on the quadratic approximation (Arora et al., 2022; Damian et al., 2022; Cohen et al., 2024).

Despite these advances, existing empirical observations and theoretical analyses remain limited in several important aspects: (i) Most studies focus on full-batch training, whereas stochastic training is more common in practice; (ii) Most studies focus on failures of the quadratic approximation caused by large positive Hessian eigenvalues, even though negative Hessian eigenvalues can also induce a similar gap between theory and practice, as we will elaborate in this work; (iii) Empirical studies are largely limited to relatively small-scale settings such as CIFAR-10 and ImageNet, leaving open whether the same failure modes persist in large-scale settings such as LLM pretraining.

**Our Contributions.** We present a systematic study of the failure modes of the quadratic approximation in stochastic, adaptive, and large-scale training, with particular focus on LLM pretraining.

- We identify and decouple a distinct failure mode of the quadratic approximation arising from small negative Hessian eigenvalues that persist during training. In this regime, the quadratic approximation predicts divergence, while the loss of LLMs continues to decrease. We term this phenomenon the *Edge of Convexity* (EoC) (Figure 1a and Section 2.2).
- We extend the definition of EoS to incorporate the effects of stochastic training, adaptive optimizers, and the EoC phenomenon. In the full-batch setting, our definition recovers the original EoS definition (Figure 1b and Section 2).
- We design efficient methods to detect both EoC and EoS at scale with theoretical guarantees (Appendix F). Our results on LLM pretraining suggest that (1) EoC consistently arises; (2) EoS is prevalent but not universal, and disappears when the LR becomes sufficiently small (e.g., after decay) or when the batch size falls below a critical threshold that scales linearly with the critical batch size (Appendix E).

## 2 MAIN RESULTS

In this section, we derive the definition of Edge of Convexity (EoC) and Edge of Stability (EoS) and introduce practical tests for identifying them. We mainly use SGD as an example to demonstrate our EoC and EoS test methods. Our framework, however, naturally extends to widely used adaptive optimizers such as Adam and its variants with appropriate modifications. The major difference is that we consider and report the *preconditioned Hessian and its spectrum* for adaptive optimizers. This extension is validated by our LLM experiments in Appendix E and further discussed in Appendix H.2. Detailed derivations of the main results are provided in Appendix H, with complete proofs included thereafter.

### 2.1 STOCHASTIC STABILITY ANALYSIS

Compared to the full-batch setting, mini-batch sampling introduces additional variance that affects stability beyond what $\lambda_{\max}$ alone can capture. To analyze this, for any fixed checkpoint $\boldsymbol{w}_{t_0}$, we consider a local quadratic approximation with stochastic Hessians:

$$Q_{\mathcal{B}_t}(\boldsymbol{w}) \coloneqq \mathcal{L}_{\mathcal{B}_t}(\boldsymbol{w}_{t_0}) + (\boldsymbol{w} - \boldsymbol{w}_{t_0})^\top \nabla \mathcal{L}_{\mathcal{B}_t}(\boldsymbol{w}_{t_0}) + \tfrac{1}{2}(\boldsymbol{w} - \boldsymbol{w}_{t_0})^\top \nabla^2 \mathcal{L}_{\mathcal{B}_t}(\boldsymbol{w}_{t_0})(\boldsymbol{w} - \boldsymbol{w}_{t_0}), \quad (1)$$

where $\mathcal{B}_t$ is a mini-batch sampled at time step $t \geq t_0$. Applying SGD with LR $\eta$ to this approximation as $\boldsymbol{w}_{t+1} = \boldsymbol{w}_t - \eta \nabla_{\boldsymbol{w}} Q_{\mathcal{B}_t}(\boldsymbol{w})$ yields a linear dynamical system whose stability depends on both the population Hessian and its covariance across batches. To be specific, instability can arise in two distinct ways. (1) If $\boldsymbol{H}_{t_0}$ has negative curvature, the objective is unbounded below, and divergence can occur for any $\eta$; (2) Even in the nonnegative-curvature subspace, mini-batch noise can make the projected second moment blow up once $\eta$ is too large. We formalize both mechanisms next.

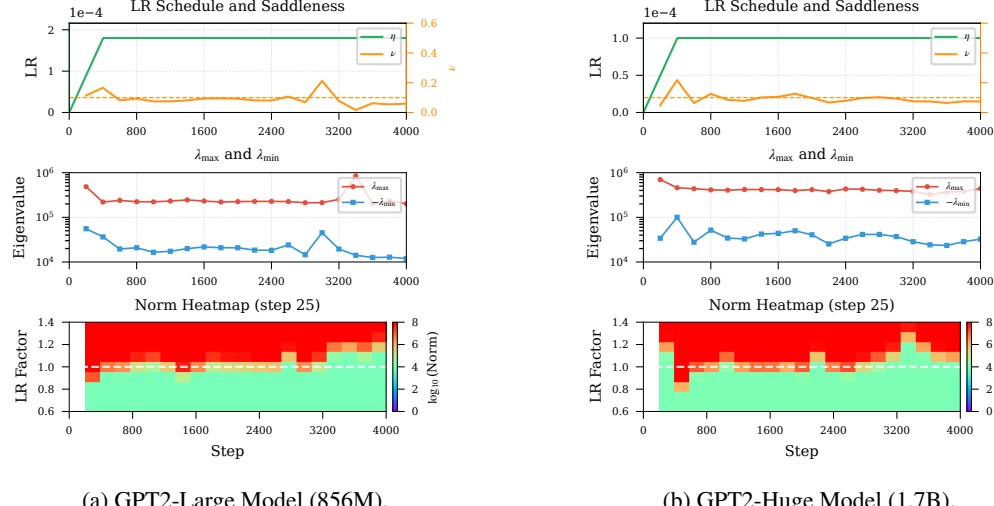

(a) GPT2-Large Model (856M).

(b) GPT2-Huge Model (1.7B).

Figure 2: **EoS and EoC tests of GPT-Large and GPT-Huge models on AdamW.** For each training run, we report: (i) the LR schedule; (ii) the top preconditioned Hessian eigenvalue $\lambda_{\max}$; (iii) the minimal preconditioned Hessian eigenvalue $\lambda_{\min}$; (iv) the saddleness $\nu$; and (v) results of the convexity-compensated local simulation, shown as the final parameter norm across nine LR factors.

**Corollary 2.1** (Negative Curvature Divergence). *Let $\boldsymbol{H}_{t_0} := \nabla^2 \mathcal{L}(\boldsymbol{w}_{t_0})$. If $\lambda_{\min}(\boldsymbol{H}_{t_0}) < 0$, then for any $\eta > 0$, the quadratic model is unbounded below, and the SGD iterates on $Q_{\mathcal{B}_t}$ diverge to $-\infty$.*

This corollary isolates instability caused purely by negative curvature, independent of $\eta$. In the remainder, we factor out these directions and focus on the locally convex subspace, where instability arises from mini-batch noise and admits a critical LR.

**Definition 2.2** (Critical LR for SGD). *Let $\boldsymbol{H}_{t_0} := \nabla^2 \mathcal{L}(\boldsymbol{w}_{t_0})$, we denote $\Pi_{t_0}^+$ as the orthogonal projection onto its eigenspace of the nonnegative eigenvalues. We define*

$$\eta_\star(\boldsymbol{w}_{t_0}) := \sup\{\eta > 0 : \sup_t \mathbb{E}\left[\|\Pi_{t_0}^+(\boldsymbol{w}_t)\|^2\right] < \infty\}.$$

Definition 2.2 considers the SGD trajectory projected onto the nonnegative-curvature subspace of $\boldsymbol{H}_{t_0}$, isolating the effect of EoC. In this restricted setting, the dynamics mirror full-batch EoS behavior and admit a critical LR. The following corollary formalizes this intuition.

**Corollary 2.3** (Positive Curvature Divergence). *Let $\boldsymbol{H}_{t_0} := \nabla^2 \mathcal{L}(\boldsymbol{w}_{t_0})$ and let $\eta_\star(\boldsymbol{w}_{t_0})$ be defined in Definition 2.2, then the SGD trajectory is unbounded above in projected second moment $\mathbb{E}\|\Pi_{t_0}^+(\boldsymbol{w}_t)\|^2$ diverges to $+\infty$ when $\eta > \eta_\star(\boldsymbol{w}_{t_0})$.*

For neural network (NN) training, once we replace the actual loss with $Q_{\mathcal{B}_t}(\boldsymbol{w})$, it is typically observed that the linear system formed by quadratic approximation diverges quickly, while the true dynamics of the NN remain stable (Figure 1a, Figure 1b).

So a natural question is: which regime leads to the failure of quadratic approximation in NN training, negative or negative curvature divergence? We identify: NN training sits on an *edge* where either divergence mechanism is *just at the threshold* of breaking the local quadratic approximation.

For the first negative divergence regime, we first define a quantity termed saddleness.

**Definition 2.4** (Saddleness). *Given any parameter point $\boldsymbol{w}_{t_0}$ whose population Hessian $\boldsymbol{H}_{t_0}$ has minimal, maximum eigenvalue denoted by $\lambda_{\min}$ and $\lambda_{\max}$. We define the saddleness $\nu$ at this point as*

$$\nu(\boldsymbol{w}_{t_0}) = \frac{\max\{0, -\lambda_{\min}\}}{\lambda_{\max}}.$$

Intuitively, $\nu(\boldsymbol{w}_{t_0})$ measures how "saddle-like" the local curvature is, by comparing the most negative curvature to the strongest positive curvature. If $\boldsymbol{H}_{t_0} \succeq 0$ (locally convex), then $\nu(\boldsymbol{w}_{t_0}) = 0$. A larger $\nu$ means the negative curvature is comparable to $\lambda_{\max}$ (strong saddle structure), while a small $\nu$ means the negative curvature exists but is weak relative to the dominant positive directions.

One important observation is: For LLM pre-training, we observe that $\nu \leq 0.1$ after the warm-up phase, which leads to the first edge area that LLM pre-training is in

> **Edge of Convexity (EoC).** Negative eigenvalues persist during training, but $\nu \approx 0$, and true dynamics remains stable, while the quadratic approximation is unbounded below, and the iterates diverge to $-\infty$ regardless of the choice of LR.

Even after removing the effect of EoC (i.e., restricting to the positive-curvature subspace in Definition 2.2), the quadratic approximation can still diverge. This divergence is caused by positive curvature when the learning rate exceeds the critical threshold $\eta_\star(\boldsymbol{w}_{t_0})$ (see Corollary 2.1). More subtly, neural network training often operates near this boundary: a tiny positive increase in the learning rate can already trigger divergence.

> **Edge of Stability (EoS) at $\boldsymbol{w}_{t_0}$.** The current LR $\eta_{t_0} \approx \eta_\star(\boldsymbol{w}_{t_0})$, a small positive perturbation of $\eta$ leads the local quadratic approximation $Q_{\mathcal{B}_t}(\boldsymbol{w})$ to $+\infty$.

One can notice that our definition recovers the full-batch EoS condition $\lambda_{\max} \approx 2/\eta$ in Cohen et al. (2021). In the full-batch case, $\nabla^2 \mathcal{L}_{\mathcal{B}_t}(\boldsymbol{w}_{t_0}) \equiv \boldsymbol{H}_{t_0}$ and the update on the locally convex subspace reduces to $\boldsymbol{x}_{t+1} = (\boldsymbol{I} - \eta\boldsymbol{H}_{t_0})\boldsymbol{x}_t$. Hence stability is equivalent to $\rho(\boldsymbol{I} - \eta\boldsymbol{H}_{t_0}) < 1$, i.e., $0 < \eta < 2/\lambda_{\max}(\boldsymbol{H}_{t_0})$, so $\eta_\star(\boldsymbol{w}_{t_0}) = 2/\lambda_{\max}(\boldsymbol{H}_{t_0})$.

In the following subsections, we propose practical methods to detect each regime that simulates local trajectories while avoiding explicit eigendecomposition.

## 2.2 EoC Test via Minimal Eigenvalue Estimation

At a checkpoint $\boldsymbol{w}_{t_0}$, EoC corresponds to persistent but small negative curvature, i.e., $\lambda_{\min}(\boldsymbol{H}_{t_0}) < 0$ and $\nu \approx 0$, under which the quadratic approximation is unbounded below and fails regardless of the LR. Our goal is to estimate $\lambda_{\min}(\boldsymbol{H}_{t_0})$ using only stochastic mini-batch access, without costly eigendecomposition.

**Stochastic Power Iteration.** We propose a stochastic power iteration method with bisection to estimate the smallest eigenvalue of a large matrix from noisy observations, without costly eigendecomposition. The key idea is to probe the stability of a shifted stochastic iteration: for a shift coefficient $\lambda$, we run a few steps of a power iteration driven by the sampled matrices and track the growth of the iterate norm. When $\lambda$ is smaller than the critical value, the dynamics admit an expansive mode and $\|\boldsymbol{x}_t\|$ rapidly blows up; once $\lambda$ is large enough (with a suitably small step size), the iteration becomes stable and the iterates remain bounded. This sharp "explode or stable" behavior enables a bisection search over $\lambda$. The full procedure is given in Algorithm 1, and its finite error bound is provided in Theorem F.1; empirical EoC results are shown in Figures 2 and 7.

**EoC Test Framework.** We use the above method to estimate the minimal eigenvalue of $\boldsymbol{H}_{t_0}$ for any chosen checkpoint $\boldsymbol{w}_{t_0}$ and use Lanczos (Lanczos, 1950) to estimate the maximum eigenvalue. Combining these two gives the estimated $\widehat{\nu}$. When $\widehat{\nu} < \nu_{\text{tol}}$, we identify the current point is at EoC. The detailed description for this framework is provided in Algorithm 2. We observe the consistent negative eigenvalues occur across various training setups. See Appendix E for more discussion.

## 2.3 EoS Test via Convexity-Compensated Local Simulation

**Compensated local loss.** We propose an efficient method to test EoS at scale when the saddleness is small. In detail, we define the compensated local loss as:

$$
\begin{aligned}
G_t(\boldsymbol{w}) :=\ & \mathcal{L}_{\mathcal{B}_t}(\boldsymbol{w}_{t_0}) + (\boldsymbol{w} - \boldsymbol{w}_{t_0})^\top \nabla \mathcal{L}_{\mathcal{B}_t}(\boldsymbol{w}_{t_0}) \\
& + \tfrac{1}{2}(\boldsymbol{w} - \boldsymbol{w}_{t_0})^\top \nabla^2 \mathcal{L}_{\mathcal{B}_t}(\boldsymbol{w}_{t_0})(\boldsymbol{w} - \boldsymbol{w}_{t_0}) + \tfrac{\lambda_{\mathsf{P}}}{2}\|\boldsymbol{w} - \boldsymbol{w}_{t_0}\|^2,
\end{aligned}
\tag{2}
$$

where $\lambda_{\mathsf{P}} \geq 0$ is called the *compensation coefficient*. We first set $\lambda_{\mathsf{P}} \geq |\lambda_{\min}|$. Likewise, we extend the critical LR in Definition 2.2 with the compensation coefficient $\lambda_{\mathsf{P}}$ as $\eta_\star^{\lambda_{\mathsf{P}}}(\boldsymbol{w}_{t_0})$ (See Equation (14) for rigorous definition). Since the saddleness is observed to be small, we can prove that $\eta_\star^{\lambda_{\mathsf{P}}}(\boldsymbol{w}_{t_0})$ is close to $\eta_\star(\boldsymbol{w}_{t_0})$ (Theorem F.2). Hence, the phase transition in the projected space spanned by positive eigenvectors can also be observed directly in this compensated local dynamics.

**EoS Test Framework.** We summarize the above discussion into a unified framework for the detection of EoS. We set the copensation coefficient using the estimation by Algorithm 1 proposed in Section 2.2 as $\lambda_{\mathsf{P}} = \widehat{\lambda}$. Then , we fix a single mini-batch sequence $\{\mathcal{B}_t\}_{t=0}^{T-1}$, for all LR factor $\alpha \in \{\alpha_1 < \cdots < \alpha_K\}$, and simulate $\boldsymbol{w}_{t+1}^{(\alpha)} = \boldsymbol{w}_t^{(\alpha)} - \alpha\eta \nabla G_t(\boldsymbol{w}_t)$, $\boldsymbol{w}_0^{(\alpha)} = 0$, as summarized in Algorithm 3. We claim a reference point $\boldsymbol{w}_{t_0}$ is in EoS, if there exists a phase transition in the norms $\|\boldsymbol{w}_T^\alpha\|$ (see the norm heat maps in Figure 2 and Figure 1b).

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

## A    CONCLUSION

This paper extends the definition of EoS to the LLM training under a unified framework, together with another clearly clarified phenomenon EoC, accounting for the two failure modes of the quadratic approximation.

Several directions remain open for future study. (i) Our experiments mainly focus to SGD and Adamw. It is interesting to see how would EoS would behave when training with other optimizers like Muon. (ii) Fully understanding EoS and EoC also calls for solid theoretical analyses of the realistic neural network training dynamics.

## B    ROADMAP OF APPENDIX

Appendix C provides a detailed discussion of related work. In Appendix D, we introduce the necessary notations and background of full-batch Edge of Stability used in the following empirical and theoretical derivations. In Appendix E, we present our detailed experimental results, together with all the figures, and Appendix G further clarifies and compensates the experimental details. Appendix F shows the theoretical guarantees for the EoC and EoS tests in practice. Appendix H and the following sections together give a self-contained derivation of our main results of EoC and EoS, with proofs and algorithm descriptions provided.

## C    RELATED WORK

**Edge of Stability.**    Early studies connected sharpness and stability to optimizer behavior in neural network training: Wu et al. (2018) analyzed SGD minima selection through dynamical stability using sharpness and non-uniformity, and Xing et al. (2018) observed SGD "bouncing" between valley walls during training. Jastrzębski et al. (2018) then showed that SGD visits increasingly sharp regions up to a maximum set by LR and batch size, and that the step is often too large along the sharpest directions. Cohen et al. (2021) later demonstrated that full-batch gradient descent typically operates at an Edge of Stability (EoS), where the largest Hessian eigenvalue oscillates near $2/\eta$ while loss decreases non-monotonically. Cohen et al. (2022) extended this phenomenon to adaptive methods via saturation of a preconditioned stability threshold. And theoretical explanations for EoS were discussed in simplified models or realistic assumptions(Arora et al., 2022; Li et al., 2022; Damian et al., 2022).

**Comparison with Andreyev & Beneventano (2024).**    The recent work (Andreyev & Beneventano, 2024) is among the most relevant to this work, which defines Edge of Stochastic Stability for SGD via the batch sharpness, where Batch Sharpness hovers around $2/\eta$. Compared to Andreyev & Beneventano (2024), our work (1) In the full-batch setting, our definition for EoS can be reduced to the original definition of EoS by Cohen et al. (2021; 2022), while theirs cannot; (2) We also identify another failure mode of the quadratic approximation called Edge of Convexity (EoC); (3) We provide scalable EoS and EoC test methods with theoretical guarantees; (4) We conduct experiments on large-scale LLM pretraining with AdamW, while most of their experiments are limited to ResNets on CIFAR-10 and SVHN with vanilla SGD.

**Training Stability of Neural Networks**    Series works studied the training stability of constant-learning-rate SGD in quadratic losses via a linear system analysis of the second moment of the parameters (Wu et al., 2018; Ma & Ying, 2021; Granziol et al., 2022; Wu et al., 2022; Mulayoff & Michaeli, 2024). They obtained tight conditions in the form of $\|\mathbb{E}[(\boldsymbol{I} - \eta\boldsymbol{H}_t)^{\otimes 2}]\| \leq 1$, which are valid instability criteria for a particular Lyapunov function. However, in the training neural networks with (S)GD, the unstable convergence phenomenon was observed (Wu et al., 2018; Xing et al., 2018; Jastrzębski et al., 2018), formally identified and termed *Edge of Stability* (EoS) (Cohen et al., 2021). Subsequently, several studies sought to explain this unstable convergence (Ma et al., 2022; Ahn et al., 2022; Wu et al., 2023; Arora et al., 2022; Damian et al., 2022; Cohen et al., 2024). Among them, Cohen et al. (2024) gave a general characterization of the oscillatory dynamics of deep learning in the EoS regime via a time-averaged differential equation termed *central flow*. However, the effectiveness of the quadratic approximation was never carefully studied in the large-scale neural network training, hence it is not clear when and how the quadratic approximation fails or not in the realistic regime.

**Hessian Spectrum of Neural Networks.** Most studies on the Hessian spectrum of neural networks reported that the Hessian spectra of neural networks consist of a "bulk" together with a few large "outliers" in magnitude (LeCun et al., 2002; Dauphin et al., 2014; Sagun et al., 2016; 2017; Ghorbani et al., 2019; Chaudhari et al., 2019; Yao et al., 2020). Among them, Sagun et al. (2016; 2017) mentioned that there are still negative eigenvalues during training, but their magnitude is much smaller than the large outliers. More recent works reported and explained a near-block diagonal structure in the Hessian spectrum of CNNs and Transformers (Zhang et al., 2024; Dong et al., 2025). However, one thing that has long been overlooked is the characterization of the negative eigenvalues in the population Hessian during large-scale training. In this work, we present detailed empirical findings of the negative eigenvalues across different setups in large-scale neural network training.

## D PRELIMINARY

### D.1 NOTATION

Throughout, $\|\cdot\|$ denotes the spectral norm for matrices and the $\ell_2$ norm for vectors. For a symmetric matrix $\boldsymbol{A}$, let $\lambda_{\max}(\boldsymbol{A})$ and $\lambda_{\min}(\boldsymbol{A})$ be its largest and smallest eigenvalues, We write $\mathrm{vec}(\cdot)$ for vectorization and $\otimes$ for the Kronecker product; $\mathrm{Cov}(\boldsymbol{M}_t)$ denotes the covariance of $\mathrm{vec}(\boldsymbol{M}_t)$. For adaptive methods, we use momentum $\boldsymbol{m}_t$, a preconditioner $\boldsymbol{P}_t$ updated by a map $\Psi$, and weight decay coefficient $\lambda_{\mathrm{WD}}$; the relevant curvature is the spectrum of $\boldsymbol{P}_t^{-1}\nabla^2\mathcal{L}(\boldsymbol{w}_t)$, equivalently that of its symmetric similarity transform $\boldsymbol{P}_t^{-1/2}\nabla^2\mathcal{L}(\boldsymbol{w}_t)\boldsymbol{P}_t^{-1/2}$. We use $\mathbb{S}^d$ to denote the set of $d \times d$ symmetric matrices and $\mathbb{S}^d_{++}$ to narrow the range to symmetric positive definite matrices. For a positive semidefinite matrix $\boldsymbol{S}$, define the seminorm $\|\boldsymbol{u}\|_{\boldsymbol{S}} := \sqrt{\boldsymbol{u}^\top \boldsymbol{S}\, \boldsymbol{u}}$.

### D.2 PROBLEM SETUP

We consider the training dynamics of a neural network $f(\boldsymbol{x}; \boldsymbol{w})$, where $\boldsymbol{x} \in \mathbb{R}^d$ denotes the input and $\boldsymbol{w} \in \mathbb{R}^p$ denotes the trainable parameter. Given a dataset $\{(\boldsymbol{x}_i, y_i)\}_{i=1}^n$ and a per-sample loss function $\ell$, the full-batch loss and mini-batch loss are defined as

$$\mathcal{L}(\boldsymbol{w}) := \frac{1}{n} \sum_{i=1}^n \ell\big(f(\boldsymbol{x}_i; \boldsymbol{w}), y_i\big),$$

$$\mathcal{L}_{\mathcal{B}}(\boldsymbol{w}) := \frac{1}{|\mathcal{B}|} \sum_{i \in \mathcal{B}} \ell\big(f(\boldsymbol{x}_i; \boldsymbol{w}), y_i\big),$$

where $\mathcal{B} \subseteq [n]$ denotes the mini-batch of size $|\mathcal{B}|$. Let $\nabla\mathcal{L}(\boldsymbol{w})$ and $\nabla^2\mathcal{L}(\boldsymbol{w})$ denote the full-batch gradient and Hessian, respectively.

### D.3 FULL-BATCH EDGE OF STABILITY

Consider full-batch gradient descent with LR $\eta$:

$$\boldsymbol{w}_{t+1} = \boldsymbol{w}_t - \eta\nabla\mathcal{L}(\boldsymbol{w}_t).$$

In a recent work, Cohen et al. (2021) observed that during neural network training, the largest eigenvalue of the Hessian, denoted by $\lambda_{\max}$, typically increases over time and eventually approaches the threshold $2/\eta$. After reaching this value, $\lambda_{\max}$ oscillates around it while the loss decreases non-monotonically. This regime is referred to as the *Edge of Stability* (EoS).

To understand why this behavior is surprising, consider the second-order Taylor expansion of the loss around a reference point $\boldsymbol{w}_{t_0}$:

$$Q(\boldsymbol{w}) := \mathcal{L}(\boldsymbol{w}_{t_0}) + (\boldsymbol{w} - \boldsymbol{w}_{t_0})^\top \nabla\mathcal{L}(\boldsymbol{w}_{t_0})$$
$$+ \tfrac{1}{2}(\boldsymbol{w} - \boldsymbol{w}_{t_0})^\top \nabla^2\mathcal{L}(\boldsymbol{w}_{t_0})(\boldsymbol{w} - \boldsymbol{w}_{t_0}).$$

Applying gradient descent to this quadratic objective yields the linear dynamical system

$$(\boldsymbol{w}_{t+1} - \boldsymbol{w}_\star) = \big(I - \eta\nabla^2\mathcal{L}(\boldsymbol{w}_{t_0})\big)(\boldsymbol{w}_t - \boldsymbol{w}_\star),$$

where $\boldsymbol{w}_\star$ satisfies $\nabla\mathcal{L}(\boldsymbol{w}_{t_0}) + \nabla^2\mathcal{L}(\boldsymbol{w}_{t_0})\boldsymbol{w}_\star = 0$.

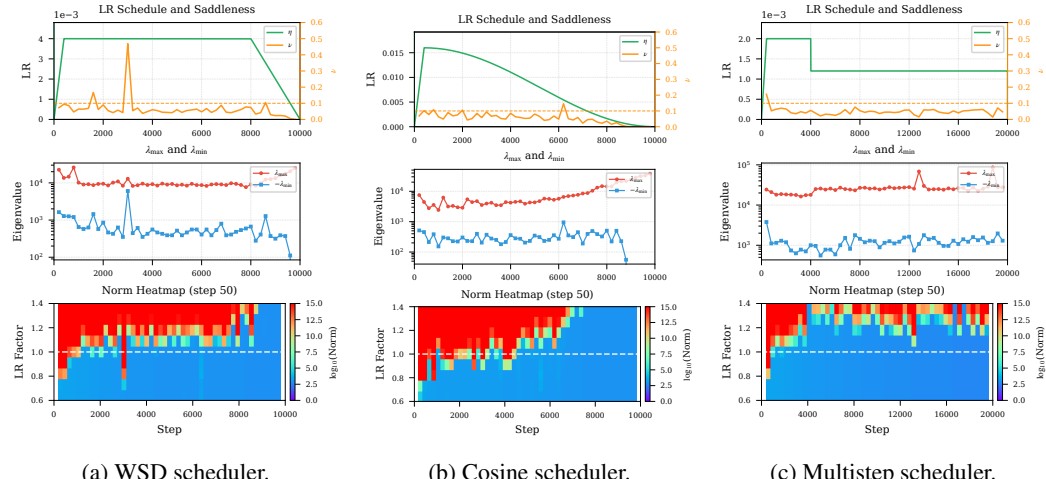

| (a) WSD scheduler. | (b) Cosine scheduler. | (c) Multistep scheduler. |

Figure 3: **EoS and EoC test under different LR schedulers.** We train a 43M GPT-like model on FineWeb-10B using AdamW with different LR schedulers. For WSD and Cosine scheduler, we tune the peak LR via a separate sweep. All runs use 400 linear warmup steps and decay the LR to zero unless otherwise specified. (A) WSD: 400 warmup / 7600 stable / 2000 decay steps, peak LR = 0.004. (B) Cosine: peak LR = 0.016. (C) Multistep: peak LR = 0.002, LR drop at step 4000 to 0.0012, no decay phase.

Classical analysis of this system shows that gradient descent is stable if and only if the Hessian is positive semidefinite and the LR is smaller than a critical threshold as $\eta \leq 2/\lambda_{\max}$.

From this perspective, the observation that $\lambda_{\max}$ concentrates near $2/\eta$ throughout training is striking. It places training dynamics precisely at the boundary of stability predicted by the quadratic model, rather than safely within it. The EoS phenomenon therefore, exposes a fundamental mismatch between the predictions of quadratic approximation and the actual behavior of neural network training.

# E EXPERIMENTS

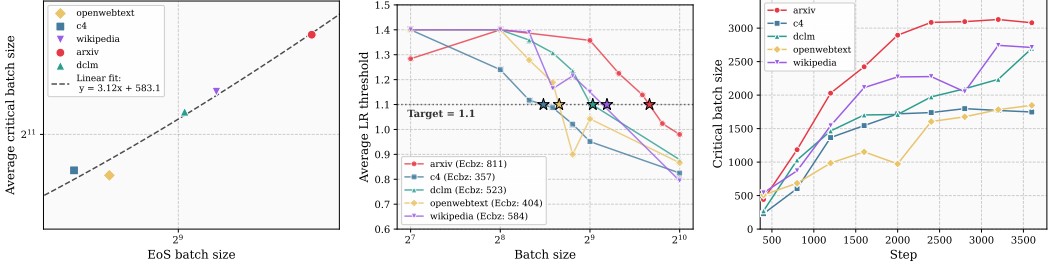

(a) EoS batch size vs. critical batch size.
(b) Average critical LR ratio vs. batch size.
(c) Critical batch size vs. checkpoints.

Figure 4: **Relationship between the EoS batch-size and the critical batch size.** (a) EoS batch size versus critical batch size across settings. (b) Estimated EoS batch-size threshold as a function of $\eta_\star^{\lambda_p}/\eta$. We use the target ratio $\eta_\star^{\lambda_p}/\eta = 1.1$. (c) Critical batch-size estimates across checkpoints; we report the average over the last 4 checkpoints as the final estimate.

## E.1 EXPERIMENTAL SETUP.

We pretrain a family of GPT-style models with size ranging from 43M to 1.7B parameters (GPT-extra-small to GPT-Huge). Unless otherwise specified, we train on FineWeb (Penedo et al., 2024) for 5B tokens (following the Modded GPT setup (Jordan et al., 2024)) with AdamW (WD= 0.1,

$\beta_1 = 0.9$), global batch size 512, and the Warm–Stable–Decay (WSD) schedule (Hu et al., 2024) for 10,000 steps (400 warmup / 7,600 stable / 2,000 linear decay to zero), excepted for GPT-extra-small to GPT-Huge models, for which we run a shorted horizon of 4000 steps with 400 warmup steps due to the compute and time limit. While standard pretraining practice dictates scaling dataset size with model capacity, our primary objective is analyzing training dynamics rather than achieving optimal convergence. Because the Edge of Stability (EoS) phenomenon consistently manifests early in the stable phase of training, this 4,000-step horizon is sufficient to demonstrate the EoS behavior while accommodating compute limits. For the LLM training runs using AdamW optimizers, all the maximum/minimal eigenvalue in experiment results are calculated on the preconditioned Hessian, as detailed in Appendix H.2. We additionally repeat key experiments on ArXiv (Clement et al., 2019), OpenWebText (Gokaslan & Cohen, 2019), C4 (Raffel et al., 2020), DCLM (Li et al., 2024), and Wikipedia (Wikimedia Foundation, 2024) to verify robustness across data sources. Full architectural specifications, HVP/simulation implementation details, and additional training/runtime settings are provided in Appendix G.

### E.2 LARGE-SCALE EoS AND EoC VERIFICATION

**EoS and EoC are prevalent across model scales** We examine the presence of Edge of Stability (EoS) and Edge of Convexity (EoC) across model scales ranging from 43M to 1.7B parameters, as shown in Figure 7 (130M, 391M), Figure 3a (43M), and Figure 2 (856M, 1.7B). We perform a fine-grained LR search and identify an optimal LR based on training loss for 43M model. Training near this optimal regime consistently shows that training is approximately at the critical threshold, exhibiting a clear EoS signature, while persistent negative curvature and a near-zero saddleness $\nu$ (typically less than $0.1$ in all our experiments) is always observed throughout training, suggesting a consistent EoC phenomenon in LLM pre-training. Additionally, for larger models with model size up to 1.7B, EoS is observed under standard LR choices. The qualitative behavior is consistent with that of smaller models, and EoC remains present throughout training. Overall, these results indicate that EoC and EoS are prevalent across model scales in practical training settings.

### E.3 ABLATION STUDY

We perform a series of ablation studies to examine the robustness of EoC and EoS across training settings.

**EoC is Robust across training configurations.** Across variations in peak LR (Figure 5), batch size (Figure 6), scheduler (Figure 3), and model scale (Figures 2, 3a and 7), we consistently observe the presence of EoC. Specifically, the saddleness $\nu$ remains below $0.1$ for most of the training time, despite some unstable spikes in a few checkpoints. Oh the other hand, the minimal eigenvalue of the population preconditioned Hessian can attain a non-negligible magnitude.

**EoS emerges around and below the optimal LR.** We vary the peak LR and report the corresponding EoS/EoC tests in Figure 5. When the peak LR is at or below the empirically optimal value (Figures 5a and 5b), we observe a clear EoS signature. In contrast, when the peak LR is set above the optimum (Figure 5c), the quadratic approximation becomes highly unstable.

**Schedulers shape the temporal profile of EoS.** As shown in Figure 3, the LR schedule strongly affects when and how long the EoS signature is observed. Under WSD, EoS is consistently present throughout the stable phase, but it fades during the subsequent decay phase. Under cosine decay, EoS is concentrated in the earlier part of training and disappears noticeably earlier than in WSD. Under the multistep schedule, EoS becomes less prominent after the abrupt LR drop.

**EoS appears only above a batch size threshold.** As shown in Figure 6, EoS is tightly coupled to the global batch size. Specifically, with a smaller batch size ($B = 256$; Figure 6a), the EoS signature is not observed. Once the batch size exceeds a threshold ($B = 512$ and $B = 1024$; Figures 6b and 6c), EoS emerges clearly and is observed reliably across checkpoints. We call this batch size threshold the EoS batch size. Figure 4c further shows that when the batch size is small (less than $2^8$), the EoS would not happen (The averaged LR threshold reaches the maximal factor $1.4$ in grid searching, thus no grid-searched LR factor would lead to divergence), suggesting that EoS similarly disappears when the batch size is below some thresholds for other commonly used pre training datasets other than fineweb. We then term the above batch size by the EoS batch size.

**The EoS batch size is smaller but linearly related to the critical batch size.** We examine the relationship between the EoS batch size and the classical critical batch size () in Figure 4. EoS is identified using the averaged critical LR ratio $\eta_\star^{\lambda_\mathsf{P}}(\boldsymbol{w}_{t_0})/\eta$ over checkpoints, with a fixed threshold of 1.1. Fitting this metric against batch size (Figure 4b) yields the EoS batch size. The critical batch size is estimated following Merrill et al. (2025) via local simulation and averaged over the last four checkpoints (Figure 4c). We find that the EoS batch size is consistently smaller than the critical batch size, while the two vary proportionally across datasets. Implementation details are deferred to Appendix G.4.

## F THEORETICAL GUARANTEES BEHIND THE MEASUREMENTS OF EOC AND EOS

In this section, we provide theoretical justification for the definitions and tests of EoC and EoS introduced in Appendix 2. A self-contained derivation that unifies the empirical procedures from Appendix 2 with the theory developed here is presented in Appendix H.

### F.1 ERROR BOUND FOR STOCHASTIC POWER ITERATION

**Theorem F.1** (informal). *Let $\boldsymbol{A} \in \mathbb{S}^d$ and $\boldsymbol{A}_t = \boldsymbol{A} + \Delta_t$ be noisy symmetric observations with $\mathbb{E}[\Delta_t] = \boldsymbol{0}$ and $|\Delta_t| \leq \sigma$ almost surely. Given a nontrivial initialization overlap with the minimal eigenvector of $\boldsymbol{A}$, Algorithm 1 with $\eta \in [\omega(1/T), , o(1/\sqrt{T})]$, horizon $T$, and $N$ bisection rounds outputs an estimate $\widehat{\lambda}_{\min}$ such that, with high probability,*

$$\left|\widehat{\lambda}_{\min} - \lambda_{\min}(\boldsymbol{A})\right| = \widetilde{\mathcal{O}}\left(\frac{1}{\eta T} + \frac{\sigma\sqrt{\log d}}{\sqrt{T}}\right) + \mathcal{O}\left(2^{-N}\right).$$

This theorem provides a finite-sample error bound for our stochastic power iteration estimator of $\lambda_{\min}$ in Appendix 2.2. It guarantees that Algorithm 1 returns a reliable estimate of $|\lambda_{\min}|$. A formal statement is deferred to Appendix H.1.2.

### F.2 EOS TEST ERROR FOR SGD

**Theorem F.2.** *Given $\mathsf{Cov}(\nabla^2 \mathcal{L}_{\mathcal{B}_t}(\boldsymbol{w}_{t_0})) \preceq \sigma^2 \boldsymbol{I}$ and the vanilla SGD, we have*

$$\frac{\eta_\star^{\lambda_\mathsf{P}}}{\eta_\star^0} \in \left[\frac{\lambda_{\max}(\lambda_{\max} + \lambda_\mathsf{P})}{(\lambda_{\max} + \lambda_\mathsf{P})^2 + \sigma^2}, \frac{\lambda_{\max}^2 + \sigma^2}{\lambda_{\max}(\lambda_{\max} + \lambda_\mathsf{P})}\right].$$

*Specifically, if $\lambda_\mathsf{P} = -\lambda_{\min}$, then we have*

$$\frac{\eta_\star^{\lambda_\mathsf{P}}}{\eta_\star^0} \in \left[\frac{\kappa^2 + \kappa}{(\kappa + 1)^2 + \alpha}, \frac{\kappa^2 + \alpha}{\kappa^2 + \kappa}\right]$$

*where $\alpha = \frac{\sigma^2}{|\lambda_{\min}|^2}$ , $\kappa := 1/\nu = \lambda_{\max}/|\lambda_{\min}|$.*

This theorem quantifies how much the critical LR (hence the EoS location) changes after adding the convexity-compensation term. In particular, when we choose $\lambda_\mathsf{P} = -\lambda_{\min}$ to cancel negative curvature, the ratio $\eta_\star^{\lambda_\mathsf{P}}/\eta_\star^0$ is controlled by the condition number $\kappa$ and the normalized noise level $\alpha = \sigma^2/|\lambda_{\min}|^2$. Therefore, our "remove EoC then test EoS" procedure does not arbitrarily shift the EoS threshold; its distortion is explicitly bounded by $\nu$ and $\alpha$.

## G DETAILED EXPERIMENTS

### G.1 EXPERIMENTAL SETUPS

**Model architectures.** We use the following models in our experiments.

We further use GPT-2 vocabulary for all models except GPT-extra-small (which uses an 8k vocabulary for fast ablations), and apply QK-normalization (Henry et al., 2020) and attention gating (Qiu et al., 2025) for stability.

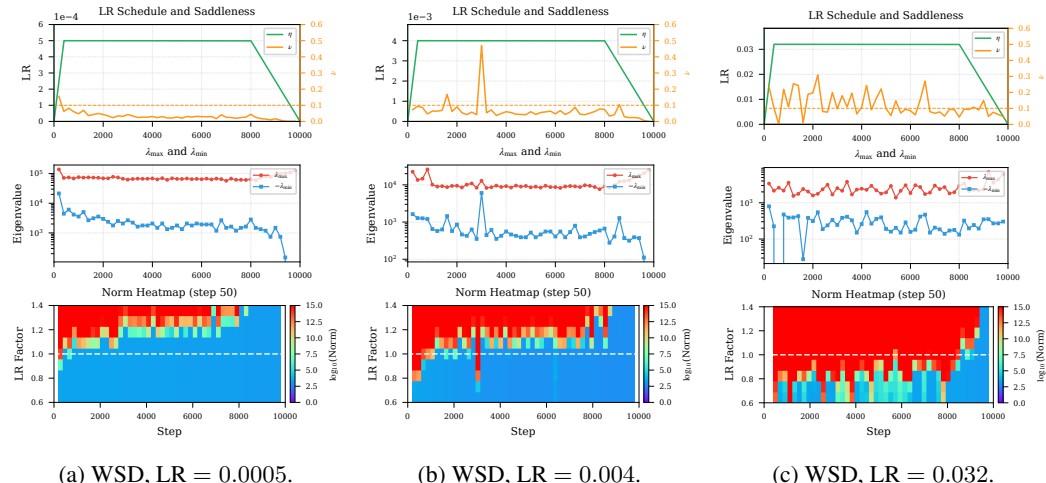

| (a) WSD, LR = 0.0005. | (b) WSD, LR = 0.004. | (c) WSD, LR = 0.032. |

Figure 5: **EoS and EoC measurements under different LRs.** We train a 43M GPT-like model on FineWeb-10B using AdamW with the WSD scheduler and vary the LR. All runs use 400 linear warmup steps, 7600 stable steps, and 2000 linear decay steps. We identify $0.004$ as the optimal peak LR via a seperate sweep.

Table 1: Model architectures used in our experiments.

| Model | #Layers | Hidden Dim | #Heads | Head Dim | Vocab Size | Params |
|---|---|---|---|---|---|---|
| GPT-extra-small | 4 | 768 | 12 | 64 | 8,192 | 43M |
| GPT-small | 12 | 768 | 12 | 64 | 50,257 | 130M |
| GPT-medium | 24 | 1,024 | 16 | 64 | 50,257 | 391M |
| GPT-large | 36 | 1,280 | 20 | 64 | 50,257 | 856M |
| GPT-huge | 48 | 1,600 | 25 | 64 | 50,257 | 1.7B |

To accelerate the computation, we run most experiments on GPT-extra-small. Unless otherwise specified, for GPT-extra-small we use 100 iterations for stochastic power iteration and 50 iterations for convexity-compensated local simulation. For larger models (GPT-small/medium/large/huge), we use 50 iterations for stochastic power iteration and 25 iterations for convexity-compensated local simulation.

We log results at a fixed set of checkpoints. For GPT-extra-small, we evaluate every 200 checkpoints, yielding 50 checkpoints from step 0 to 10,000. For GPT-small and GPT-medium, we evaluate every 400 checkpoints, yielding 25 checkpoints from step 0 to 10,000. For GPT-large and GPT-huge, due to computational constraints, we evaluate every 200 steps from step 0 to 4,000.

For adaptive optimizers like AdamW, the max/min eigenvalues are calculated on preconditioned Hessian.

**HVP computation.** We compute Hessian–vector products (HVPs) using PyTorch's autograd by differentiating a gradient–vector inner product:

$$\boldsymbol{g}(\theta) = \nabla_\theta \mathcal{L}(\theta), \qquad \mathrm{HVP}(\boldsymbol{v}) = \nabla_\theta\big(\boldsymbol{g}(\theta)^\top \boldsymbol{v}\big) = \nabla_\theta^2 \mathcal{L}(\theta)\,\boldsymbol{v}. \tag{3}$$

Below we estimate its compute cost at the level of a single *linear layer*; this is representative because transformers are composed predominantly of linear operators (attention projections and MLP projections), and their runtime is largely dominated by GEMMs.

**Per-layer GEMM counting (linear layer).** Consider a linear layer

$$\boldsymbol{y} = \boldsymbol{x}\boldsymbol{W}, \tag{4}$$

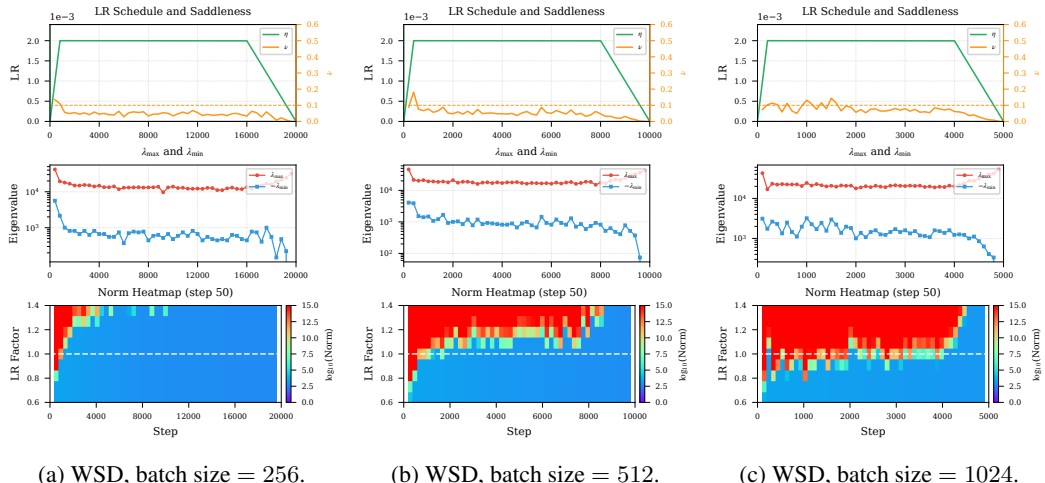

(a) WSD, batch size $= 256$.  (b) WSD, batch size $= 512$.  (c) WSD, batch size $= 1024$.

Figure 6: **EoS and EoC measurements under different batch sizes.** We train a 43M GPT-like model on FineWeb-10B using AdamW with the WSD scheduler and vary the global batch size, while fixing the LR to $0.002$. (A)–(C) Batch sizes $= 256, 512,$ and $1024$, respectively.

where $\boldsymbol{x} \in \mathbb{R}^{B \times d_{\text{in}}}$, $\boldsymbol{W} \in \mathbb{R}^{d_{\text{in}} \times d_{\text{out}}}$, and $\boldsymbol{y} \in \mathbb{R}^{B \times d_{\text{out}}}$. Let $\boldsymbol{g} = \partial \mathcal{L} / \partial \boldsymbol{y}$ be the upstream gradient. The forward pass uses one GEMM:

$$\boldsymbol{y} = \boldsymbol{x}\boldsymbol{W} \quad \Rightarrow \quad 1 \text{ GEMM}. \tag{5}$$

The first-order backward pass computes gradients w.r.t. $\boldsymbol{x}$ and $\boldsymbol{W}$:

$$\frac{\partial \mathcal{L}}{\partial \boldsymbol{x}} = \boldsymbol{g}\boldsymbol{W}^\top, \qquad \frac{\partial \mathcal{L}}{\partial \boldsymbol{W}} = \boldsymbol{x}^\top \boldsymbol{g}, \tag{6}$$

which costs two GEMMs. Hence one forward+backward through a linear layer costs $\approx 3$ GEMMs (ignoring elementwise ops).

The above autograd recipe computes an HVP by effectively backpropagating through the gradient computation (a second reverse-mode pass). For matmul-dominated layers, this incurs a constant-factor overhead comparable to an additional "forward+backward" through the same linear operators, plus extra GEMMs from differentiating the backward equations. Concretely, augmenting the forward/backward with the required second-order terms yields approximately

$$\underbrace{1}_{\text{forward}} + \underbrace{2}_{\text{backward}} + \underbrace{2}_{\text{differentiate backward (w.r.t. } \boldsymbol{W})} + \underbrace{4}_{\text{propagate directional adjoints}} \approx 9 \text{ GEMMs}, \tag{7}$$

i.e., about $3\times$ **the cost of one gradient evaluation** (9 vs. 3 GEMMs per linear layer), or equivalently $9\times$ **the cost of a forward pass** in the GEMM-dominated regime. Since transformer blocks are largely composed of such linear layers (QKV projections, attention output projection, and the two MLP projections), this constant-factor overhead provides a practical estimate of HVP cost for our models.

**Single simulation.** A single EoS simulation under a fixed training setting consists of the following procedure.

- **Step 1 (pretraining and checkpointing).** We pretrain the model under the specified setting (architecture, dataset, optimizer, scheduler, batch size, etc.). During training, we save checkpoints at a fixed frequency. Each checkpoint will be analyzed independently in the local-simulation stage below.

- **Step 2 (estimating the top eigenvalue).** For each checkpoint, we estimate the population top eigenvalue $\lambda_{\max}$ of the Hessian. Concretely, we form an empirical *average Hessian* by averaging Hessian-vector products over a fixed set of mini-batches, and run the Lanczos algorithm on this averaged operator to obtain an estimate of $\lambda_{\max}$.

- **Step 3 (locating the convexity compensation).** For each checkpoint, we run Algorithm 1 to estimate the *optimal convexity compensation* $\lambda_{\text{P}}$. Intuitively, $\lambda_{\text{P}}$ is the smallest additional

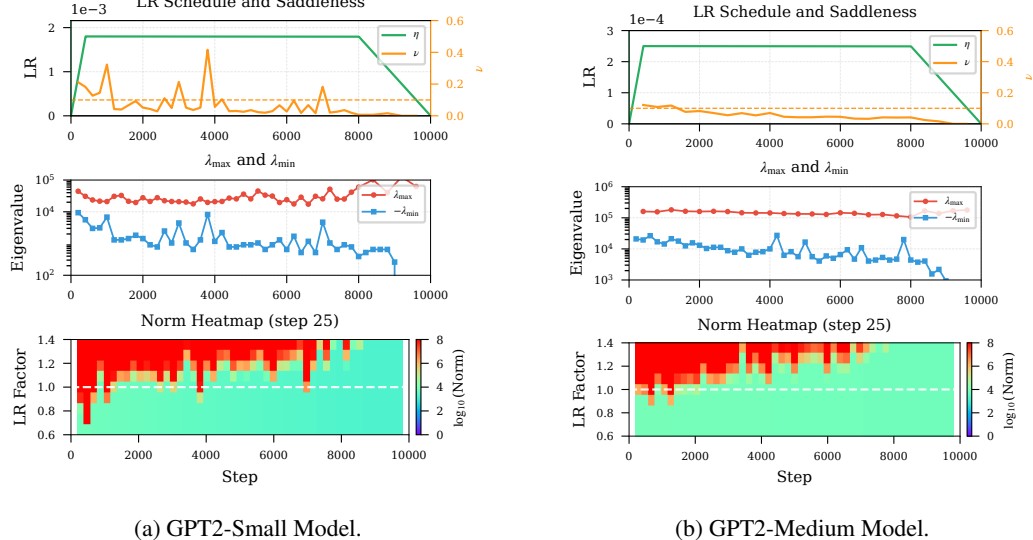

(a) GPT2-Small Model.

(b) GPT2-Medium Model.

Figure 7: **EoS and EoC Tests of GPT-Small and GPT-Medium models.**

regularization (in our local model) that removes the dominant negative-curvature effect at this checkpoint. We use this value as an operational measure of the strength of EoC.

- **Step 4 (LR-factor sweep and EoS heatmaps).** For each checkpoint, we run the local simulation on the modified approximate loss (with the estimated $\lambda_P$) across a fixed grid of learning-rate factors $\{0.6, 0.8, \ldots, 1.3, 1.4\}$. We record the final parameter norm produced by each run and visualize the results as EoS heatmaps. This sweep provides a coarse but stable picture of the local stability boundary.

- **Step 5 (estimating the critical LR).** For each checkpoint, we run Algorithm 5 to estimate the critical LR $\eta_\star^{\lambda_P}$ under convexity compensation $\lambda_P$. We compare the ratio $\eta_\star^{\lambda_P}/\eta$ with the theoretical quantity $\kappa/(1+\kappa)$. Here $\eta$ is the original LR.

**Operational criteria.** In practice, we declare a checkpoint to be in an EoS phase if the estimated ratio $\eta_\star^{\lambda_P}/\eta_\star$ falls within a tolerance band (we use $[0.9, 1.1]$). Separately, we declare a checkpoint to exhibit EoC if the estimated convexity compensation $\lambda_P$ is consistently bounded away from zero. We provide a full example report of this simulation pipeline in Figure 11.

### G.2    RESULTS OF GPT-SMALL AND GPT-MEDIUM

Figure 7 shows the EoS/EoC measurements for GPT-small and GPT-medium. For both model sizes, we observe a consistent EoC phenomenon across training: the estimated convexity compensation $\lambda_P$ stays reliably bounded away from zero over a wide range of checkpoints. In addition, the EoS signature remains visible through a large portion of training. Overall, the qualitative behaviors observed in GPT-extra-small persist for GPT-small and GPT-medium.

### G.3    ROBUSTNESS OF THE CONVEXITY COMPENSATION

Here is an example to show the robustness of the convexity compensation. Even if the convexity compensation is a little above the threshold, we can still observe the sharp phase transition.

### G.4    DETAILED EOS BATCH-SIZE EXPERIMENT SETTING

We study five datasets (ArXiv, C4, DCLM, Wikipedia, and OpenWebText) using GPT-extra-small. We fix the optimizer (AdamW with $WD = 0.1$) and use the WSD scheduler throughout. For each dataset, we first estimate the critical batch size (CBZ) using the local-simulation method of Merrill et al. (2025).

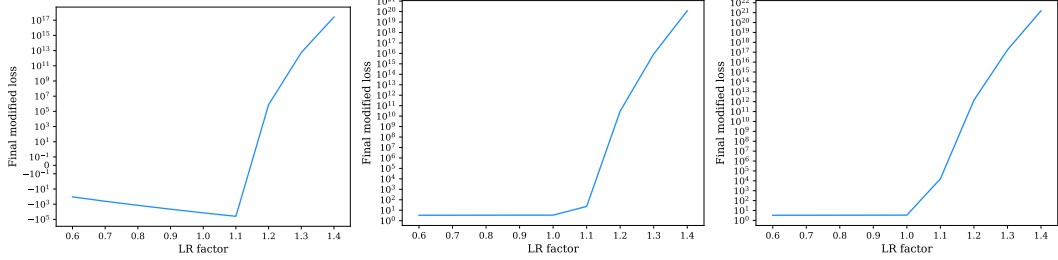

(a) No convexity compensation (= 0).    (b) Optimal compensation ($\approx 408$).    (c) Compensation $= 600$.

Figure 8: **Effects of different convexity compensations.** We train a 43M GPT-like model with AdamW on FineWeb-10B and run EoS tests with different convexity compensation values and report loss on local step 25: (a) 0 (no compensation), (b) $\approx 408$ (optimal), and (c) 600.

To vary the batch size, we follow the scaling rules in Malladi et al. (2024); Marek et al. (2025): when increasing the batch size from $B$ to $kB$, we keep all hyperparameters fixed except we set

$$\beta_2 \leftarrow \beta_2^k, \qquad \eta \leftarrow \sqrt{k}\,\eta.$$

For each batch size $B$, we estimate the EoS diagnostic ratio $\eta_\star^{\lambda_P}/\eta_\star^0$, where $\eta_\star^0$ denotes the critical LR without compensation. We then interpolate this ratio as a function of $B$ and define the *EoS batch size* (EBZ) as the smallest $B$ for which the ratio reaches $1.1$.

Figure 4a compares EBZ to CBZ. Across datasets, EBZ is consistently smaller than CBZ, while the two quantities are strongly correlated. Empirically, we observe an approximately linear relation, with CBZ $\approx 3 \times$ EBZ in our settings.

### G.5 ABLATION STUDY ON LR ABRUPT DROP

We evaluate two-step scheduler variants in Figure 9. We find that the drop ratio strongly affects whether EoS re-emerges after the drop. In particular, when the drop ratio is sufficiently small (typically below $0.5$ in our experiments), the EoS signature returns quickly and remains stable. Within the range of peak LRs tested in Figure 9, this re-emergence behavior is qualitatively robust.

### G.6 ABLATION STUDY ON LR DECAY FOR EoC

**EoC vanishes in LR decay.** We further find that EoC is influenced by LR decay, illustrated in Figure 3. As the LR becomes sufficiently small, the minimum eigenvalue gradually approaches zero, suggesting that EoC diminishes only in the small-LR regime.

### G.7 ABLATION STUDY ON LR LINEAR DECAY

We evaluate different linear-decay configurations in Figure 10. Consistent with the two-step ablations, the *decay ratio* plays a primary role in determining whether and when EoS re-emerges during the decay phase. Moreover, the transition time depends on the decay length: longer decay schedules lead to a slower and more gradual transition, roughly scaling with the number of decay steps in our settings.

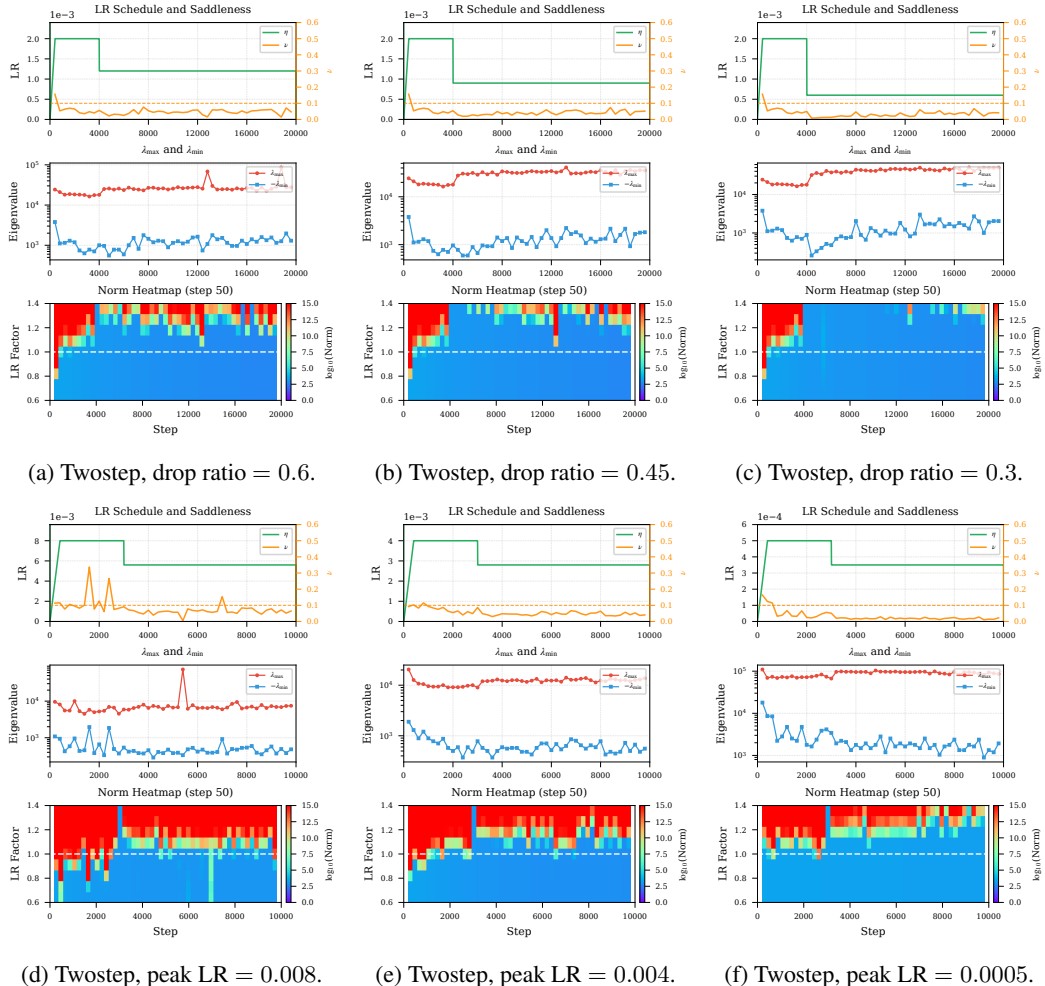

(a) Twostep, drop ratio = 0.6.  (b) Twostep, drop ratio = 0.45.  (c) Twostep, drop ratio = 0.3.

(d) Twostep, peak LR = 0.008.  (e) Twostep, peak LR = 0.004.  (f) Twostep, peak LR = 0.0005.

Figure 9: **Ablations for the Peak LR of abrupt drop.** We train a 43M GPT-like model and vary the peak LR and the drop rate of the two-step scheduler. (a)–(c) Peak LR is fixed at $0.002$, with drop rates $0.6$, $0.45$, and $0.3$; the drop step is 4000. (d)–(f) Drop rate is fixed at $0.7$, with peak LR values $0.008$, $0.004$, and $0.0005$; the drop step is 3000.

## H  DETAILED DERIVATION OF EOC AND EOS

In this part, we reorganize our main empirical methods for the measurement of EoC & EoS and the corresponding theoretical analysis throughout the empirical methods.

### H.1  VANILLA SGD

Fix a training checkpoint $\boldsymbol{w}_{t_0}$. We study the training dynamics through a stochastic quadratic surrogate built from mini-batches, which induces a linear stochastic dynamical system under SGD. This viewpoint reveals two distinct ways the quadratic model can fail: (i) *negative-curvature instability* due to $\lambda_{\min}(\nabla^2 \mathcal{L}(\boldsymbol{w}_{t_0})) < 0$ (Edge of Convexity, EoC), and (ii) *positive-curvature instability* when the LR exceeds a critical threshold (Edge of Stability, EoS). Since these two instabilities are coupled in practice, we further introduce a convexity-compensated surrogate that suppresses the negative-curvature instability while preserving the positive-curvature stability threshold, enabling a scalable local characterization of EoS.

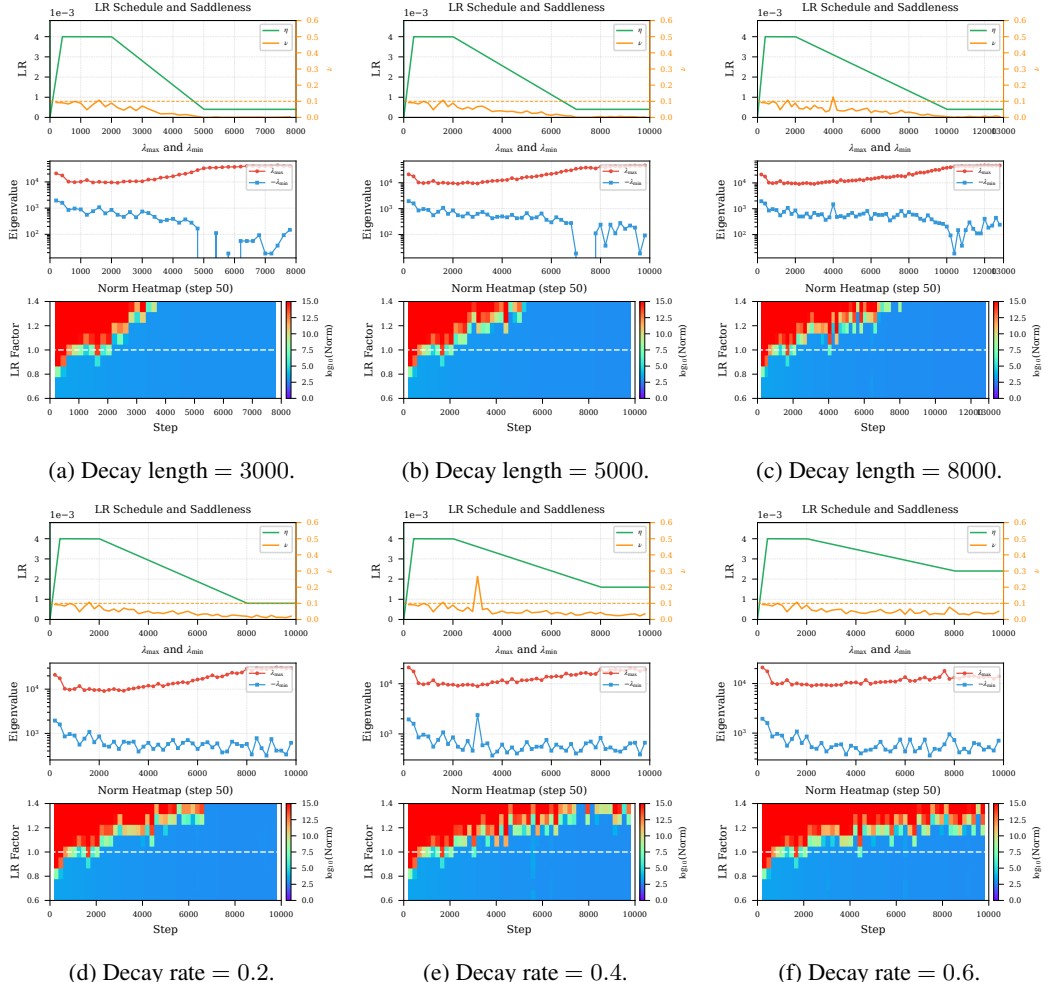

(a) Decay length $= 3000$.      (b) Decay length $= 5000$.      (c) Decay length $= 8000$.

(d) Decay rate $= 0.2$.      (e) Decay rate $= 0.4$.      (f) Decay rate $= 0.6$.

Figure 10: **Ablations for the linear decay phase.** We train a 43M GPT-like model and vary the decay length and the decay rate of the linear decay phase. (a)–(c) The decay rate is fixed at $0.1$, with decay lengths of 3000, 5000, and 8000 steps. (d)–(f) The decay length is fixed at 6000 steps, with decay rates of $0.2$, $0.4$, and $0.6$.

### H.1.1 STOCHASTIC STABILITY ANALYSIS

For stochastic optimization, mini-batch sampling introduces additional variance that affects stability beyond what $\lambda_{\max}$ alone can capture. To analyze this, for any fixed checkpoint $\boldsymbol{w}_{t_0}$ during training, we consider a local quadratic approximation with stochastic Hessians:

$$
\begin{aligned}
Q_{\mathcal{B}_t}(\boldsymbol{w}) := \mathcal{L}_{\mathcal{B}_t}(\boldsymbol{w}_{t_0}) + (\boldsymbol{w} - \boldsymbol{w}_{t_0})^\top \nabla \mathcal{L}_{\mathcal{B}_t}(\boldsymbol{w}_{t_0}) \\
+ \tfrac{1}{2}(\boldsymbol{w} - \boldsymbol{w}_{t_0})^\top \nabla^2 \mathcal{L}_{\mathcal{B}_t}(\boldsymbol{w}_{t_0})(\boldsymbol{w} - \boldsymbol{w}_{t_0}),
\end{aligned}
\tag{8}
$$

where $\mathcal{B}_t$ is a mini-batch sampled at time step $t \geq t_0$. Applying SGD with LR $\eta$ to this approximation as

$$
\boldsymbol{w}_{t+1} = \boldsymbol{w}_t - \eta \nabla_{\boldsymbol{w}} Q_{\mathcal{B}_t}(\boldsymbol{w}),
\tag{9}
$$

which yields a linear dynamical system whose stability depends on both the expected Hessian and its covariance across batches. The proof of Theorem H.1 is given in Appendix I.

**Theorem H.1** (Linear System for SGD). *Assume the batches are i.i.d., and there exists a $\boldsymbol{w}_\star$ such that:*

$$
\nabla \mathcal{L}(\boldsymbol{w}_{t_0}) + \nabla^2 \mathcal{L}(\boldsymbol{w}_{t_0})(\boldsymbol{w}_\star - \boldsymbol{w}_{t_0}) = 0.
\tag{10}
$$

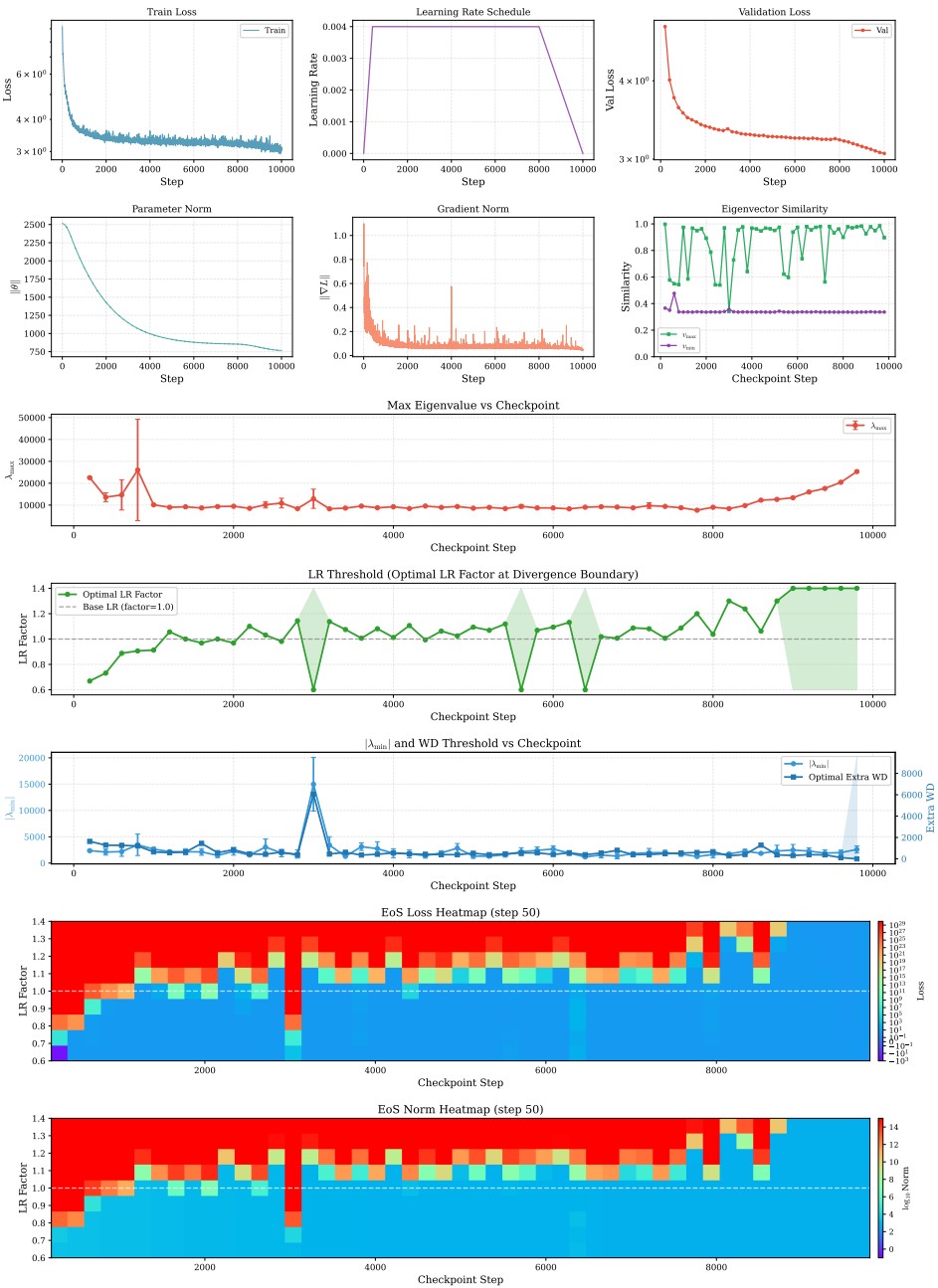

Figure 11: **Summary of a representative training run.** We show training dynamics for GPT-extra-small trained on FineWeb-10B using AdamW with LR $0.004$, weight decay $0.1$, and the WSD learning-rate scheduler.

*If we apply SGD to $Q_{\mathcal{B}_t}(\boldsymbol{w})$ as (9), the evolution of the first and second moments of the parameter trajectory $\boldsymbol{w}_t$,*

$$\boldsymbol{m}_t^{(1)} := \mathbb{E}[\boldsymbol{w}_t - \boldsymbol{w}_\star],$$

$$\boldsymbol{m}_t^{(2)} := \text{vec}\Big(\mathbb{E}[(\boldsymbol{w}_t - \boldsymbol{w}_\star)(\boldsymbol{w}_t - \boldsymbol{w}_\star)^\top]\Big),$$

*is given by:*

$$\begin{bmatrix} \boldsymbol{m}_{t+1}^{(1)} \\ \boldsymbol{m}_{t+1}^{(2)} \end{bmatrix} = \begin{bmatrix} \mathbb{E}[\boldsymbol{M}_t], & 0 \\ \boldsymbol{S}, & \mathbb{E}[\boldsymbol{M}_t \otimes \boldsymbol{M}_t] \end{bmatrix} \begin{bmatrix} \boldsymbol{m}_t^{(1)} \\ \boldsymbol{m}_t^{(2)} \end{bmatrix} + \begin{bmatrix} \boldsymbol{0} \\ \boldsymbol{b} \end{bmatrix}. \tag{11}$$

*where*

$$\boldsymbol{M}_t := I - \eta \nabla^2 \mathcal{L}_{\mathcal{B}_t}(\boldsymbol{w}_{t_0}),$$

*and $\boldsymbol{S}$ and $\boldsymbol{b}$ are constants depending on the batch distribution, $\boldsymbol{w}_{t_0}$, and $\boldsymbol{w}_\star$.*

The stability of this linear system is governed by $\|\mathbb{E}[\boldsymbol{M}_t]\|$ and $\|\mathbb{E}[\boldsymbol{M}_t \otimes \boldsymbol{M}_t]\|$. The first controls the mean trajectory, while the second controls the variance. The following corollaries give the stability conditions; see Appendix J for the proof.

**Corollary H.2** (Negative Curvature Divergence). *Let $\boldsymbol{H}_{t_0} := \nabla^2 \mathcal{L}(\boldsymbol{w}_{t_0})$. If $\lambda_{\min}(\boldsymbol{H}_{t_0}) < 0$, then for any $\eta > 0$, the quadratic model is unbounded below, and the SGD iterates on $Q_{\mathcal{B}_t}$ diverge to $-\infty$.*

The above corollary accounts for the impact of strictly negative curvature, which would lead the quadratic objective to decrease without bound along the direction of the negative eigenvalues. After that, the surrogate iterates run away, and the quadratic loss diverges to $-\infty$.

As we mentioned, this failure mode of negative curvature is separated from the instability that is typically induced by $\eta$. To study the latter, we factor out the negative-curvature directions and focus on the subspace where the quadratic model is locally convex and define the corresponding critical LR.

**Definition H.3** (Critical LR for SGD). *Let $\boldsymbol{H}_{t_0} := \nabla^2 \mathcal{L}(\boldsymbol{w}_{t_0})$, we denote $\Pi_{t_0}^+$ as the orthogonal projection onto its eigenspace of the nonnegative eigenvalues. We define*

$$\eta_\star(\boldsymbol{w}_{t_0}) := \sup\{\eta > 0 : \sup_t \mathbb{E}\|\Pi_{t_0}^+(\boldsymbol{w}_t - \boldsymbol{w}_\star)\|^2 < \infty\}.$$

Definition H.3 is intentionally projected onto the nonnegative-curvature subspace of $\boldsymbol{H}_{t_0}$. It is therefore well-defined even when $\lambda_{\min}(\boldsymbol{H}_{t_0}) < 0$ and should be interpreted as the critical LR threshold for the positive-curvature instability. The following corollary shows this intuition clearly.

**Corollary H.4** (Positive Curvature Divergence). *Let $\boldsymbol{H}_{t_0} := \nabla^2 \mathcal{L}(\boldsymbol{w}_{t_0})$ and let $\eta^\star(\boldsymbol{w}_{t_0})$ be defined in Definition 2.2, then the SGD trajectory is unbounded above in projected second moment $\mathbb{E}\|\Pi_{t_0}^+(\boldsymbol{w}_t - \boldsymbol{w}_\star)\|^2$ diverges to $+\infty$ when $\eta > \eta^\star(\boldsymbol{w}_{t_0})$.*

For neural network training, once we replace the actual loss with $Q_{\mathcal{B}_t}(\boldsymbol{w})$, it is typically observed that the linear system formed by quadratic approximation diverges quickly, while the true dynamics of the neural network remain stable ( Figure 1a, Figure 1b).

So a natural question is: which regime is the neural network training in, negative curvature divergence or negative curvature divergence? We identify: it sits on an *edge* where either divergence mechanism is *just at the threshold* of breaking the local quadratic approximation.

For the first negative divergence regime, we first define a quantity termed saddleness.

**Definition H.5.** *Given any parameter point $\boldsymbol{w}_{t_0}$ whose population Hessian $\boldsymbol{H}_{t_0}$ has minimal, maximum eigenvalue denoted by $\lambda_{\min}$ and $\lambda_{\max}$. We define the saddleness $\nu$ at this point as*

$$\nu(\boldsymbol{w}_{t_0}) = \frac{\max\{0, -\lambda_{\min}\}}{\lambda_{\max}}.$$

Intuitively, $\nu(\boldsymbol{w}_{t_0})$ measures how "saddle-like" the local curvature is, by comparing the magnitude of the most negative curvature to the strongest positive curvature. If $\boldsymbol{H}_{t_0} \succeq 0$ (locally convex), then

$\nu(\boldsymbol{w}_{t_0}) = 0$. A larger $\nu$ means the negative curvature is comparable to $\lambda_{\max}$ (strong saddle structure), while a small $\nu$ means the negative curvature exists but is weak relative to the dominant positive directions.

One important observation is: For LLM pre-training and image classification, for all steps $t$ after the warmup phase, we observe that $\nu \leq 0.1$, which leads to the first edge area that neural network training is in

> **Edge of Convexity (EoC).** Negative eigenvalues persist during training, but $\nu \approx 0$, and true dynamics remains stable, while the quadratic approximation is unbounded below, and the iterates diverge to $-\infty$ regardless of the choice of LR.

And for the positive divergence regime, we identify that the neural network training is at the following edge area

> **Edge of Stability (EoS) at $\boldsymbol{w}_{t_0}$.** The current LR $\eta_{t_0} \approx \eta_\star(\boldsymbol{w}_{t_0})$, a small positive perturbation of $\eta$ leads the local quadratic approximation $Q_{\mathcal{B}_t}(\boldsymbol{w})$ to $+\infty$.

In the full-batch (deterministic) case, $\nabla^2 \mathcal{L}_{\mathcal{B}_t}(\boldsymbol{w}_{t_0}) \equiv \boldsymbol{H}_{t_0}$ and the update on the locally convex subspace reduces to $\boldsymbol{x}_{t+1} = (\boldsymbol{I} - \eta \boldsymbol{H}_{t_0})\boldsymbol{x}_t$. Hence stability is equivalent to $\rho(\boldsymbol{I} - \eta \boldsymbol{H}_{t_0}) < 1$, i.e., $0 < \eta < 2/\lambda_{\max}(\boldsymbol{H}_{t_0})$, so $\eta_\star(\boldsymbol{w}_{t_0}) = 2/\lambda_{\max}(\boldsymbol{H}_{t_0})$. Therefore our criterion $\eta_{t_0} \approx \eta_\star(\boldsymbol{w}_{t_0})$ recovers the classical EoS condition $\eta\lambda_{\max} \approx 2$ in Cohen et al. (2021).

These two edge areas are geometrically distinct but coupled in the training dynamics. A naive approach is to decompose the parameter space $\mathbb{R}^p$ into orthogonal subspaces: $U_+$ spanned by eigenvectors with positive eigenvalues, and $U_-$ spanned by eigenvectors with non-positive eigenvalues. One could then project the trajectory onto $U_+$ and $U_-$ to detect each mode. However, computing this eigendecomposition is prohibitive for high-dimensional neural networks. This leads to our first challenge: *How to separate the effect of EoS and EoC in quadratic approximation?*

Even if we can separate EoS and EoC, unlike the small-scale full-batch setting—where the EoS can be characterized via $\eta$ and $2/\lambda_{\max}$—the stability of quadratic approximation for large-scale stochastic training is determined by:

$$\mathbb{E}[\boldsymbol{M}_t \otimes \boldsymbol{M}_t] = \mathbb{E}[\boldsymbol{M}_t] \otimes \mathbb{E}[\boldsymbol{M}_t] + \mathsf{Cov}(\boldsymbol{M}_t),$$

where $\boldsymbol{M}_t := \boldsymbol{I} - \eta\nabla^2 \mathcal{L}_{\mathcal{B}_t}(\boldsymbol{w}_t)$ and $\mathsf{Cov}(\boldsymbol{M}_t)$ is the covariance of $\mathsf{vec}(\boldsymbol{M}_t)$ induced by mini-batch sampling. The computation is prohibitive due to two reasons:

Furthermore, unlike the full-batch case where $\eta_\star = 2/\lambda_{\max}$, we cannot give an explicit expression for $\eta_\star$ in the stochastic setting for two reasons:

**(i) Batch noise shifts the stability threshold.** The second-moment stability depends on $\|\mathbb{E}[\boldsymbol{M}_t \otimes \boldsymbol{M}_t]\|$, which decomposes as:

$$\mathbb{E}[\boldsymbol{M}_t \otimes \boldsymbol{M}_t] = \mathbb{E}[\boldsymbol{M}_t] \otimes \mathbb{E}[\boldsymbol{M}_t] + \mathsf{Cov}(\boldsymbol{M}_t),$$

where $\mathsf{Cov}(\boldsymbol{M}_t)$ is the covariance of $\mathsf{vec}(\boldsymbol{M}_t)$ induced by mini-batch sampling. Without the covariance term, the spectral radius would be $\max_i(1 - \eta\lambda_i)^2$, yielding $\eta_\star = 2/\lambda_{\max}$. However, $\mathsf{Cov}(\boldsymbol{M}_t)$ can shift the spectral radius significantly, causing $\eta_\star$ to deviate from $2/\lambda_{\max}$.

**(ii) The true $\lambda_{\max}$ is computationally inaccessible.** Even if the threshold were $2/\lambda_{\max}$, computing $\lambda_{\max}$ of the full-batch Hessian $\nabla^2 \mathcal{L}(\boldsymbol{w})$ is infeasible for large-scale training, as it requires access to the entire dataset (or population). While one can estimate $\lambda_{\max}$ from mini-batch Hessians, these estimates have high variance and may not reflect the true full-batch spectrum.

Hence, our second challenge is, *how to measure EoS without estimation of $\eta_\star$?*

These challenges motivate our simulation-based approach, which directly probes the stability boundary without requiring explicit knowledge of $\lambda_{\max}$ or $\mathsf{Cov}(\boldsymbol{M}_t)$.

In the following subsections, we propose practical methods to detect each regime by simulating local trajectories, bypassing explicit eigendecomposition.

At a checkpoint $\boldsymbol{w}_{t_0}$, EoC corresponds to persistent negative population curvature, i.e., $\lambda_{\min}(\nabla^2 \mathcal{L}(\boldsymbol{w}_{t_0})) < 0$, under which the quadratic approximation is unbounded below and fails regardless of the LR. Our goal is to estimate $\lambda_{\min}(\nabla^2 \mathcal{L}(\boldsymbol{w}_{t_0}))$ using only stochastic mini-batch access, without explicit eigendecomposition.

**Stochastic Power Iteration Bisection.**   To this end, we provide a stochastic power iteration bisection method that can estimate the smallest eigenvalue of a large matrix $\boldsymbol{A}$ without costly eigendecomposition. Specifically, given a matrix $\boldsymbol{A} \in \mathbb{R}^{d \times d}$, and we get an sequence of independent random observations $\{\boldsymbol{A}_t\}_{t=1}^{T}$ with $\mathbb{E}\boldsymbol{A}_t = \boldsymbol{A}$ and $\mathsf{Cov}(\boldsymbol{A}_t) = \boldsymbol{\Sigma}$. We first decide a large range for the minimal eigenvalue as $|\lambda_{\min}(\boldsymbol{A})| \in [0, \Lambda]$. We then consider the following power iteration

$$\boldsymbol{x}_{t+1} = \left(\boldsymbol{I} - \eta(\boldsymbol{A}_t + \lambda\boldsymbol{I})\right)\boldsymbol{x}_t.$$

Intuitively, consider applying the above update for $T$ steps. If $\lambda < \lambda^\star$, the mean dynamics admit an expansive mode and $\|x_T\|$ explodes; if $\lambda \geq \lambda^\star$ and $\eta$ is sufficiently small, the iteration becomes stable. Algorithm 1 formalizes this intuition via an EXPLODETEST predicate and performs $N$ rounds of bisection to return the smallest stable $\widehat{\lambda}$. The high-probability accuracy of this estimator is guaranteed by the following theorem.

**Theorem H.6** (Error bound for stochastic power-iteration bisection). *Let $\boldsymbol{A} \in \mathbb{S}^d$ be a symmetric matrix with minimal eigenvalue $\lambda_{\min}$ and a unit eigenvector $\boldsymbol{u}_{\min}$. At each inner iteration $t$, we observe $\boldsymbol{A}_t = \boldsymbol{A} + \Delta_t$, where $\Delta_t \in \mathbb{S}^d$ satisfies $\mathbb{E}[\Delta_t \mid \mathcal{F}_{t-1}] = \boldsymbol{0}$, where the filtration $\mathcal{F}_t := \sigma(\Delta_0, \ldots, \Delta_{t-1})$ and $\|\Delta_t\| \leq \sigma$ almost surely.[1] Assume the initialization $\boldsymbol{x}_0$ satisfies*

$$\frac{|\boldsymbol{x}_0^\top \boldsymbol{u}_{\min}|}{\|\boldsymbol{x}_0\|_2} \geq \varepsilon \tag{12}$$

*for some absolute constant $\varepsilon > 0$.[2] Run Algorithm 1 with inner horizon $T$, bisection rounds $N$, search range $[0, \Lambda]$, and inner stepsize $\eta$ satisfying $\eta = o(1/\sqrt{T})$, $\eta = \omega(1/T)$. Let $\widehat{\lambda}_{\mathrm{shift}}$ be the estimated shift returned by Algorithm 1 and define $\widehat{\lambda}_{\min} := -\widehat{\lambda}_{\mathrm{shift}}$. Then for any $\delta \in (0,1)$, with probability at least $1 - \delta$,*

$$\left|\widehat{\lambda}_{\min} - \lambda_{\min}\right| = \tilde{\mathcal{O}}\left(\frac{1}{\eta T} + \frac{\sigma\sqrt{\log d}}{\sqrt{T}}\right) + \mathcal{O}\left(\frac{\Lambda}{2^{-N}}\right),$$

*where $\widetilde{O}(\cdot)$ hides polylogarithmic factors in $(d, 1/\delta)$ and $O(\cdot)$ hides absolute constants.*

Theorem H.6 justifies Algorithm 1 as a consistent estimator of the minimal eigenvalue. The detailed proof of this theorem is provided in Appendix M.

**EoC Test Framework.**   We now use the stochastic power iteration method to estimate the minimal eigenvalue of $\boldsymbol{H}_{t_0}$ for any chosen checkpoint $\boldsymbol{w}_{t_0}$, and we observe the consistent negative eigenvalues during each stage of training across various training setups, as discussed in Appendix E. Also, as shown in Figure 1a, once we replace the instant loss function with a local quadratic approximation and run several iterations, we see that the approximated loss sharply diverges to $-\infty$ while the real loss curve remains stable. Combining the above 2 steps for the detection of EoC gives an obvious takeaway: **EoS is always observed across the neural network training.** The detailed description for this framework is provided in Algorithm 2.

### H.1.3   EoS Test via Convexity-Compensated Local Simulation

Now we introduce our framework for the detection of EoS, which is more challenging compared with the straightforward detection of EoC. Detecting EoS requires projecting the parameter trajectory onto the subspace spanned by eigenvectors with positive eigenvalues, which is computationally prohibitive. However, we observe a significant gap between the maximum and minimal Hessian eigenvalues in practice:

---

[1]The same statement holds under i.i.d. mean-zero noise as a special case.

[2]Condition (12) can be achieved by a normal initialization $\boldsymbol{x}_0 \sim \mathcal{N}(0, \boldsymbol{I}_d)$ with high probability. We can freely generalize this theorem to the case with normal initialization, shown at the end of Appendix M.

**Observation H.7.** *For LLM pre-training, for all steps $t$ after the warmup phase, we observe that*

$$\lambda_{\max} \geq 10 \cdot |\lambda_{\min}|,$$

*i.e., saddleness $\nu \leq 0.1$. See Appendix E for empirical validations.*

This motivates us to add a convexity compensation term that eliminates negative curvature effects while preserving positive curvature instability. The idea is simple: by adding $\frac{\lambda_P}{2}\|\boldsymbol{\theta}\|^2$ to the local loss, we shift all Hessian eigenvalues upward by $\lambda_P$. If $\lambda_P \geq |\lambda_{\min}|$, the effective Hessian becomes positive semi-definite, removing negative curvature effects. Figure 8 shows that an appropriate compensation can decouple EoS from the EoC. Furthermore, since $\kappa \geq 10$ by the above observation, this shift only slightly perturbs the large positive eigenvalues that govern EoS.

To make the above idea rigorous, we first let $\boldsymbol{\theta} := \boldsymbol{w} - \boldsymbol{w}_{t_0}$ denote the displacement from the current parameters. For each mini-batch $\mathcal{B}_t$, we define the compensated local loss as:

$$\begin{aligned} G_t(\boldsymbol{\theta}) &:= \mathcal{L}_{\mathcal{B}_t}(\boldsymbol{w}_{t_0}) + \boldsymbol{\theta}^\top \nabla \mathcal{L}_{\mathcal{B}_t}(\boldsymbol{w}_{t_0}) \\ &\quad + \tfrac{1}{2}\boldsymbol{\theta}^\top \nabla^2 \mathcal{L}_{\mathcal{B}_t}(\boldsymbol{w}_{t_0})\boldsymbol{\theta} + \tfrac{\lambda_P}{2}\|\boldsymbol{\theta}\|^2, \end{aligned} \tag{13}$$

where $\lambda_P \geq 0$ is called the compensation coefficient. The derivation of this linear system and its corresponding stability analysis is provided in Appendix I and Appendix J.

Now we can extend the definition of critical LR in Definition H.3 with the compensation coefficient $\lambda_P$ as

$$\eta_\star^{\lambda_P}(\boldsymbol{w}_{t_0}) := \sup\{\eta > 0 : \sup_t \mathbb{E}\|\Pi_{t_0}^{+'}(\boldsymbol{w}_t - \boldsymbol{w}_\star)\|^2 < \infty\}, \tag{14}$$

where $\Pi_{t_0}^{+'}$ denotes the orthogonal projection onto $\boldsymbol{H}_{t_0}$'s eigenspace spanned by the eigenvectors whose eigenvalue $\mu$ satisfies $\mu + \lambda_P > 0$.

The next question is to decide how large the compensation coefficient should be. Ideally, we should set $\lambda_P$ as small as possible and make $G_t$ convex in expectation over time $t$, which gives $\lambda_P = |\lambda_{\min}|$.

In practice, we set the compensation coefficient using the estimation by Algorithm 1 proposed in Appendix H.1.2 as $\lambda_P = \widehat{\lambda}$ so that the compensated curvature $\boldsymbol{H}_{t_0} + \lambda_P \boldsymbol{I}$ is (approximately) positive semidefinite, which suppresses the EoC mode while leaving the positive-curvature stability threshold nearly unchanged. Specifically, we can show that the perturbation of the critical LR with $\lambda_P$, denoted by $\eta_\star^{\lambda_P}$ can be bounded by the condition number $\kappa$.

**Theorem H.8** (EoS Detection Error for SGD). *Given the Batch covariance matrix* $\mathsf{Cov}(\nabla^2 \mathcal{L}_{\mathcal{B}_t}(\boldsymbol{w}_{t_0})) \preceq \sigma^2 \boldsymbol{I}$, *consider the vanilla SGD, then we have*

$$\frac{\eta_\star^{\lambda_P}}{\eta_\star^0} \in \left[\frac{\lambda_{\max}(\lambda_{\max} + \lambda_P)}{(\lambda_{\max} + \lambda_P)^2 + \sigma^2}, \frac{\lambda_{\max}^2 + \sigma^2}{\lambda_{\max}(\lambda_{\max} + \lambda_P)}\right],$$

*where $\lambda_{\max}, \lambda_{\min}$ represent the maximum and minimal eigenvalues of $\nabla^2 \mathcal{L}(\boldsymbol{w}_{t_0})$. Specifically, if $\lambda_P = -\lambda_{\min}$, then we have*

$$\frac{\eta_\star^{\lambda_P}}{\eta_\star^0} \in \left[\frac{\kappa^2 + \kappa}{(\kappa + 1)^2 + \alpha}, \frac{\kappa^2 + \alpha}{\kappa^2 + \kappa}\right]$$

*where $\alpha = \frac{\sigma^2}{|\lambda_{\min}|^2}$, $\kappa := 1/\nu = \lambda_{\max}/|\lambda_{\min}|$.*

The proof of Theorem H.8 is given in Appendix K. Combining this theorem with Observation H.7, we provide a framework for detecting EoS decoupling from the impact of EoC.

**EoS Test Framework.** We summarize the above discussion into a unified framework for the detection of EoS:

- **Step1: remove the EoC mode with Convexity compensation.** When $\lambda_{\min}(\nabla^2 \mathcal{L}(\boldsymbol{w}_{t_0})) < 0$, the quadratic surrogate is unbounded below and diverges regardless of $\eta$ (EoC), which would mask the EoS mode. Using the procedure in Algorithm 1, we obtain $\lambda_P \approx -\lambda_{\min}$ and define the compensated mini-batch surrogate $G_{\mathcal{B}_t}$ with curvature $\nabla^2 \mathcal{L}_{\mathcal{B}_t}(\boldsymbol{w}_{t_0}) + \lambda_P \boldsymbol{I}$. Theorem H.8 together with Observation H.7 ensures that this compensation perturbs the positive-curvature stability threshold only by a relative factor $1 + O(1/\kappa)$.

- **Step2: probe the stability boundary by local simulation.** Fix a single mini-batch sequence $\{\mathcal{B}_t\}_{t=0}^{T-1}$ and simulate, for all LR factor $\alpha \in \{\alpha_1 < \cdots < \alpha_K\}$,

$$\boldsymbol{\theta}_{t+1}^{(\alpha)} = \boldsymbol{\theta}_t^{(\alpha)} - \alpha\eta\nabla G_t(\boldsymbol{\theta}_t), \quad \boldsymbol{\theta}_0^{(\alpha)} = 0,$$

as summarized in Algorithm 3.

- **Step3: Estimate the critical LR and decide the existence of EoS.** We declare *explosion* if $\max_{0 \leq t \leq T} \|\theta_t^{(\alpha)}\|_2 \geq M$ and estimate the critical LR $\eta_\star^{\lambda_P}$ by

$$\widehat{\eta}_\star^{\lambda_P}(\boldsymbol{w}_{t_0}) := \min\{\alpha_k\eta : \theta^{(\alpha_k)} \text{ explodes in } T \text{ steps}\},$$

with the convention that if no $\alpha_k$ explodes, we only obtain the lower bound $\widehat{\eta}_{\lambda_P}^\star(\boldsymbol{w}_{t_0}) > \alpha_K\eta$. We then quantify proximity to EoS via the ratio $r(\boldsymbol{w}_{t_0}) := \eta/\widehat{\eta}_{\lambda_P}^\star(\boldsymbol{w}_{t_0})$: When $r(\boldsymbol{w}_{t_0})$ is close to 1, the current LR is close to $\widehat{\eta}_{\lambda_P}^\star$, indicating proximity to the local EoS regime. $r \ll 1$ indicates a stable margin, and $r > 1$ indicates the compensated surrogate is already unstable at $\eta$.

**Remark H.9.** *Most training runs use a LR schedule, whereas our probe assumes a constant $\eta$. This is not a conflict: the local simulation spans only a small number of steps relative to training, during which $\eta$ changes negligibly for commonly used schedules (e.g., WSD or cosine), so we treat it as approximately constant.*

## H.2 ADAPTIVE GRADIENT METHODS

Modern neural networks, such as transformer-based large language models, are usually trained with adaptive optimizers such as Adam (Kingma, 2014) and its variants (Loshchilov & Hutter, 2017). In this section, we extend the previous definition and detection framework of EoS to a class of Adaptive optimizers. Specifically, we consider a class of Adaptive Gradient Methods (AGMs) (Li et al., 2025) with the following general form:

$$\mathbf{m}_{t+1} = \beta_1 \cdot \mathbf{m}_t + (1 - \beta_1) \cdot \nabla\mathcal{L}_{\mathcal{B}_t}(\boldsymbol{w}_t) \tag{15}$$

$$\boldsymbol{P}_{t+1} = \Psi(\boldsymbol{P}_t, \nabla\mathcal{L}_{\mathcal{B}_t}(\boldsymbol{w}_t), \boldsymbol{w}_t) \tag{16}$$

$$\boldsymbol{w}_{t+1} = \boldsymbol{w}_t - \eta\boldsymbol{P}_{t+1}^{-1}\mathbf{m}_{t+1} - \eta\lambda_{\mathrm{WD}}\boldsymbol{w}_t, \tag{17}$$

where momentum $\boldsymbol{m}_t \in \mathbb{R}^d$ and the preconditioner mapping $\Psi : \mathbb{S}_{++}^d \times \mathbb{R}^d \times \mathbb{R}^d \to S_{++}^d$. Here $\Psi$ is assumed to be a slowly-varying mapping, so we can treat $\boldsymbol{P}_t$ as fixed over a short time window. Note that AdamW, SGDM, and SGD are special cases of this AGM class. For preconditioned optimizers, the relevant spectrum is that of the preconditioned Hessian $\boldsymbol{P}_t^{-1}\nabla^2\mathcal{L}(\boldsymbol{w}_t)$; we use $\mu_{\max}$ and $\mu_{\min}$ to denote its largest and smallest eigenvalues when the context is clear.

Similar to the case for SGD, we first build the linear dynamical system for AGMs when applying local quadratic approximation with a convexity compensation term $Q_{\mathcal{B}_t}(\boldsymbol{w}_t)$ for the loss function $\mathcal{L}_{\mathcal{B}_t}(\boldsymbol{w}_t)$, and then the following stability analysis would lead us to the definitions of EoC and EoS for AGMs.

### H.2.1 EoC AND EoS FOR AGMS

Fix a checkpoint $\boldsymbol{w}_{t_0}$ and freeze the preconditioner over a short window as $\boldsymbol{P} := \boldsymbol{P}_{t_0}$. Under the same local quadratic approximation as in Appendix H.1, an AGM update can be written as a linear stochastic system in an augmented state $\boldsymbol{z}_t$ (e.g., stacking the momentum $\boldsymbol{m}$ and parameter error $\boldsymbol{w}_t - \boldsymbol{w}_\star$, where $\boldsymbol{w}_\star$ is given in (10))

$$\boldsymbol{z}_{t+1} = \boldsymbol{M}_t(\eta)\boldsymbol{z}_t + \boldsymbol{q}_t.$$

Detailed definitions and the derivation of this system can be found in Appendix L. In this system, the relevant curvature for stability is the spectrum of the preconditioned Hessian $\boldsymbol{P}^{-1}\boldsymbol{H}$ or equivalently $\widetilde{\boldsymbol{H}}_{t_0} := \boldsymbol{P}^{-1/2}\boldsymbol{H}_{t_0}\boldsymbol{P}^{-1/2}$, whose eigenvalues we denote by $\mu_1, \cdots, \mu_d$ (with $\mu_{\max}, \mu_{\min}$ as extremes) in the following derivation.

Compared with vanilla SGD, the critical LR $\eta_\star$ is accordingly defined here and all stability statements in Appendix H.1 carry over after replacing the Hessian eigenvalues $\{\lambda_i\}$ by the preconditioned eigenvalues $\{\mu_i + \lambda_{\mathrm{WD}}\}$ (incorporating weight decay as an additional isotropic shift): $\mu_i + \lambda_{\mathrm{WD}} \geq$

$0, \quad \eta \le \eta_\star, \forall i \in [d]$. Correspondingly, we can define the saddleness with the weight decay factor as $\nu_{\lambda_{\mathrm{WD}}}$.

In the same way, the above stable conditions imply the definition of EoC and EoS for AGMs:

- **Edge of Convexity (EoC) for AGMs.** When $\mu_{\min} + \lambda_{\mathrm{WD}} < 0$, but $\nu_{\lambda_{\mathrm{WD}}} \approx 0$, and the quadratic approximation is unbounded below, and the iterates diverge to $-\infty$.
- **Edge of Stability (EoS) for AGMs.** When $\eta_{t_0} \approx \eta_\star(\boldsymbol{w}_{t_0})$, a small positive perturbation of $\eta$ causes the local quadratic approximation $Q_{\mathcal{B}_t}(\boldsymbol{w})$ going to $+\infty$.

Again, by an analogous argument as in Appendix H.1.3, we add a convexity compensation term on $Q_{\mathcal{B}_t}$ to avoid the impact of negative curvature (those $\mu$ such that $\mu + \lambda_{\mathrm{WD}} < 0$) as

$$
\begin{aligned}
G_t(\boldsymbol{\theta}) := \mathcal{L}_{\mathcal{B}_t}(\boldsymbol{w}_{t_0}) + \boldsymbol{\theta}^\top \nabla \mathcal{L}_{\mathcal{B}_t}(\boldsymbol{w}_{t_0}) \\
+ \tfrac{1}{2}\boldsymbol{\theta}^\top \nabla^2 \mathcal{L}_{\mathcal{B}_t}(\boldsymbol{w}_{t_0})\boldsymbol{\theta} + \tfrac{\lambda_{\mathrm{P}}}{2}\|\boldsymbol{\theta}\|_{\boldsymbol{P}}^2,
\end{aligned}
$$

In Appendix H.1, the convexity compensation term $\frac{\lambda_P}{2}\|\theta\|_2^2$ shifts the local curvature of $\boldsymbol{H}$ by $\lambda_P I$ and here we aim to shift the spectrum of $\boldsymbol{P}^{-1}\boldsymbol{H}$, thus we replace the convexity compensation term with $\frac{\lambda_{\mathrm{P}}}{2}\|\boldsymbol{\theta}\|_{\boldsymbol{P}}^2$. And we use this proxy for the detection of EoS for AGMs.

In the same fashion of Equation (14), the critical LR with compensation $\eta_\star^{\lambda_{\mathrm{P}}}$ is defined here. Then we have the following theorem, suggesting that the impact of $\lambda_{\mathrm{P}}$ is bounded by the spectral property of the preconditioned Hessian $\boldsymbol{P}^{-1}\boldsymbol{H}$.

**Theorem H.10** (EoS Detection Error). *Given the Batch covariance matrix* $\mathsf{Cov}(\nabla^2 \mathcal{L}_{\mathcal{B}_t}(\boldsymbol{w}_{t_0})) \preceq \sigma^2 \boldsymbol{I}$, *consider AGMs in the form of Equations* (15) *to* (17) *with* $\beta_1 = 0$*, it holds that*

$$
\frac{\eta_\star^{\lambda_{\mathrm{P}}}}{\eta_\star^0} \in \left[ \frac{\mu_{\max}(\mu_{\max} + \lambda_{\mathrm{P}})}{(\mu_{\max} + \lambda_{\mathrm{P}})^2 + \sigma^2}, \frac{\mu_{\max}^2 + \sigma^2}{\mu_{\max}(\mu_{\max} + \lambda_{\mathrm{P}})} \right],
$$

*where* $\mu_{\max}, \mu_{\min}$ *represent the maximum and minimal eigenvalues of* $\boldsymbol{P}^{-1}\nabla^2 \mathcal{L}(\boldsymbol{w}_{t_0}) + \lambda_{\mathrm{WD}}\boldsymbol{I}$. *Specifically, if* $\lambda_{\mathrm{P}} = -\mu_{\min}$*, then we have*

$$
\frac{\eta_\star^{\lambda_{\mathrm{P}}}}{\eta_\star^0} \in \left[ \frac{\kappa^2 + \kappa}{(\kappa + 1)^2 + \alpha}, \frac{\kappa^2 + \alpha}{\kappa^2 + \kappa} \right]
$$

*where* $\alpha = \frac{\sigma^2}{|\mu_{\min}|}$ *,* $\kappa = \mu_{\max}/|\mu_{\min}|$.

The proof of Theorem H.10 is given in Appendix K, where Theorem K.1 itself is a restatement of the above theorem. Following the approach in Appendix H.1.3, we give the test of EoC and EoS for AGMs in Algorithm 4 and Algorithm 5.

# I  PROOF OF THEOREM H.1

*Proof.* We give the characterization of a more general system with an extra weight decay term $\frac{\lambda_P}{2}\|w - w_0\|_2^2$, which is introduced in Appendix H.1.3. One can see that the result of Theorem H.1 is a special case when $\lambda_P = 0$.

We consider optimizing the following proximal loss function:

$$
\begin{aligned}
f_t(\boldsymbol{w}) := \mathcal{L}_{\mathcal{B}_t}(\boldsymbol{w}_0) &+ (\boldsymbol{w} - \boldsymbol{w}_0)^\top \boldsymbol{g}\mathcal{L}_{\mathcal{B}_t}(\boldsymbol{w}_0) \\
&+ \tfrac{1}{2}(\boldsymbol{w} - \boldsymbol{w}_0)^\top \nabla^2 \mathcal{L}_{\mathcal{B}_t}(\boldsymbol{w}_0)(\boldsymbol{w} - \boldsymbol{w}_0) \\
&+ \tfrac{\lambda_P}{2}\|\boldsymbol{w} - \boldsymbol{w}_0\|_2^2,
\end{aligned}
$$

where we term $\lambda_P$ the compensation coefficient. For notational simplicity, we define:

- (Reparameterization) $\boldsymbol{\theta} := \boldsymbol{w} - \boldsymbol{w}_0$, $g_t(\boldsymbol{\theta}) := f_t(\boldsymbol{w})$;
- (Gradient) $\boldsymbol{g}_t := \boldsymbol{g}\mathcal{L}_{\mathcal{B}_t}(\boldsymbol{w}_0)$, $\boldsymbol{g} := \mathbb{E}_{\mathcal{B}_t}[\nabla \mathcal{L}_{\mathcal{B}_t}(\boldsymbol{w}_0)]$;
- (Hessian) $\boldsymbol{H}_t := \nabla^2 \mathcal{L}_{\mathcal{B}_t}(\boldsymbol{w}_0)$, $\boldsymbol{H} := \mathbb{E}_{\mathcal{B}_t}[\nabla^2 \mathcal{L}_{\mathcal{B}_t}(\boldsymbol{w}_0)]$;
- (Reference Loss) $\mathcal{L}_t := \mathcal{L}_{\mathcal{B}_t}(\boldsymbol{w}_0)$, $\mathcal{L} := \mathbb{E}_{\mathcal{B}_t}[\mathcal{L}_{\mathcal{B}_t}(\boldsymbol{w}_0)]$;
- (Effective Hessian) $\boldsymbol{A}_t := \boldsymbol{H}_t + \lambda_P \cdot \boldsymbol{I}$, $\boldsymbol{A} := \mathbb{E}_{\mathcal{B}_t}[\boldsymbol{H}_t + \lambda_P \cdot \boldsymbol{I}]$;
- (Contraction Map) $\boldsymbol{C}_t := I - \eta \boldsymbol{A}_t$, $\boldsymbol{C} := \mathbb{E}_{\mathcal{B}_t}[I - \eta \boldsymbol{A}_t]$.

Thus, the objective function can be rewritten as:

$$
g_t(\boldsymbol{\theta}) = \mathcal{L}_t + \boldsymbol{\theta}^\top \boldsymbol{g}_t + \frac{1}{2}\boldsymbol{\theta}^\top \boldsymbol{H}_t \boldsymbol{\theta} + \frac{\lambda_P}{2}\|\boldsymbol{\theta}\|_2^2.
$$

We analyze the dynamics of Stochastic Gradient Descent (SGD) given by:

$$
\boldsymbol{\theta}_{t+1} = \boldsymbol{\theta}_t - \eta(\boldsymbol{g}_t + \boldsymbol{A}_t \boldsymbol{\theta}_t). \tag{18}
$$

In this notation, we restate Condition (10) as: we assume the existence of an optimal parameter vector $\boldsymbol{x}_\star$ such that:

$$
\mathbb{E}[\boldsymbol{g}_t + \boldsymbol{A}_t \boldsymbol{x}_\star] = \boldsymbol{g} + \boldsymbol{A}\boldsymbol{x}_\star = 0.
$$

We now examine the deviation of the current parameter from the optimal one, defined as:

$$
\epsilon_t := \boldsymbol{\theta}_t - \boldsymbol{x}_\star.
$$

The SGD recurrence relation can be expressed in terms of the error $\epsilon_t$ as:

$$
\epsilon_{t+1} = \boldsymbol{C}_t \cdot \epsilon_t - \eta(\boldsymbol{g}_t + \boldsymbol{A}_t \boldsymbol{x}_\star).
$$

We are interested in the evolution of the first and second moments of the error $\epsilon_t$. We introduce:

$$
\begin{aligned}
\boldsymbol{m}^{(1)} &:= \mathbb{E}[\epsilon_t], \quad \boldsymbol{M}^{(2)} := \mathbb{E}[\epsilon_t \epsilon_t^\top], \\
\boldsymbol{m}^{(2)} &:= \mathsf{vec}(\boldsymbol{M}).
\end{aligned}
$$

For the first moment, we derive:

$$
\begin{aligned}
\boldsymbol{m}_{t+1}^{(1)} &= \mathbb{E}[\epsilon_{t+1}] \\
&= \mathbb{E}[\boldsymbol{\theta}_{t+1} - \boldsymbol{x}_\star] \\
&= \mathbb{E}[\boldsymbol{\theta}_t - \eta(\boldsymbol{g}_t + \boldsymbol{A}_t \boldsymbol{\theta}_t) - \boldsymbol{x}_\star] \\
&= \mathbb{E}[\epsilon_t - \eta \boldsymbol{A}_t \epsilon_t - \eta(\boldsymbol{g}_t + \boldsymbol{A}_t \boldsymbol{x}_\star)] \\
&= \mathbb{E}[\epsilon_t - \eta \boldsymbol{A}_t \epsilon_t] - \mathbb{E}[\eta(\boldsymbol{g}_t + \boldsymbol{A}_t \boldsymbol{x}_\star)] \\
&= (I - \eta \boldsymbol{A})\boldsymbol{m}_t^{(1)} \\
&= \boldsymbol{C} \cdot \boldsymbol{m}_t^{(1)}.
\end{aligned}
$$

Next, we derive the update rule for the second moment. Let $\boldsymbol{r}_t := \boldsymbol{g}_t + \boldsymbol{A}_t \boldsymbol{x}_\star$. Observe that:

$$
\begin{aligned}
\epsilon_{t+1}\epsilon_{t+1}^\top &= \left[\boldsymbol{C}_t\epsilon_t - \eta\boldsymbol{r}_t\right]\left[\boldsymbol{C}_t\epsilon_t - \eta\boldsymbol{r}_t\right]^\top \\
&= \boldsymbol{C}_t\epsilon_t\epsilon_t^\top\boldsymbol{C}_t - \eta\boldsymbol{r}_t\epsilon_t^\top\boldsymbol{C}_t \\
&\quad - \eta\boldsymbol{C}_t\epsilon_t\boldsymbol{r}_t^\top + \eta^2\boldsymbol{r}_t\boldsymbol{r}_t^\top.
\end{aligned}
$$

Recall that $\mathsf{vec}(\boldsymbol{A}\boldsymbol{X}\mathbf{m}) = (\mathbf{m}^\top \otimes \boldsymbol{A})\cdot\mathsf{vec}(\boldsymbol{X})$, where $\otimes$ denotes the Kronecker product. Additionally, we define:

$$
\begin{aligned}
\boldsymbol{b} &:= \mathsf{vec}\left(\mathbb{E}\left[\boldsymbol{r}_t\boldsymbol{r}_t^\top\right]\right), \\
\boldsymbol{S} &:= \mathbb{E}[\boldsymbol{C}_t \otimes \boldsymbol{r}_t + \boldsymbol{r}_t \otimes \boldsymbol{C}_t].
\end{aligned}
$$

Consequently, the update for the vectorized second moment is:

$$
\begin{aligned}
\boldsymbol{m}_{t+1}^{(2)} &= \mathsf{vec}(\boldsymbol{M}_{t+1}) \\
&= \mathsf{vec}(\mathbb{E}[\epsilon_{t+1}\epsilon_{t+1}^\top]) \\
&= \mathbb{E}[\boldsymbol{C}_t \otimes \boldsymbol{C}_t]\cdot\boldsymbol{m}_t^{(2)} + \eta\boldsymbol{S}\cdot\boldsymbol{m}_t^{(1)} + \eta^2\boldsymbol{b}.
\end{aligned}
$$

The first and second moments evolve according to the following linear dynamical system:

$$
\begin{bmatrix}\boldsymbol{m}_{t+1}^{(1)} \\ \boldsymbol{m}_{t+1}^{(2)}\end{bmatrix} = \begin{bmatrix}\boldsymbol{C} & \mathbf{0} \\ \eta\boldsymbol{S} & \mathbb{E}[\boldsymbol{C}_t \otimes \boldsymbol{C}_t]\end{bmatrix}\begin{bmatrix}\boldsymbol{m}_t^{(1)} \\ \boldsymbol{m}_t^{(2)}\end{bmatrix} + \begin{bmatrix}\mathbf{0} \\ \eta^2\boldsymbol{b}\end{bmatrix}. \tag{19}
$$

Equation (19) completes the proof. $\qquad\square$

## J PROOF OF COROLLARY H.4 AND COROLLARY H.2

The stability of the linear system (19) is governed by the spectral radius (maximum absolute value of eigenvalues) of the transition matrix. Since the matrix is block lower triangular, we only need to consider the maximum and minimal eigenvalues of the diagonal blocks $\boldsymbol{C}$ and $\mathbb{E}[\boldsymbol{C}_t \otimes \boldsymbol{C}_t]$. In other words, the stability is equivalent to the following condition:

$$
\max\{\rho(\boldsymbol{C}), \rho\left(\mathbb{E}[\boldsymbol{C}_t \otimes \boldsymbol{C}_t]\right)\} \leq 1,
$$

where the operator $\rho(\cdot)$ denotes the spectrum norm when applied to a matrix.

For $\boldsymbol{C}$, we have $\boldsymbol{C} = I - \eta\boldsymbol{A} = I - \eta(\boldsymbol{H} + \lambda_\mathsf{P}\cdot\boldsymbol{I})$. Assuming the eigenvalues of $\boldsymbol{H}$ are $\lambda_1, \lambda_2, \ldots, \lambda_d$, the maximum eigenvalue of $\boldsymbol{C}$ is denoted by $c_{\max} = \max_i(1 - \eta\lambda_i - \eta\lambda_\mathsf{P})$, and the minimal eigenvalue is denoted by $c_{\min} = \min_i(1 - \eta\lambda_i - \eta\lambda_\mathsf{P})$. By the condition $\rho(\boldsymbol{C}) \leq 1$ We have

$$
c_{\max} \leq 1, \quad c_{\min} \geq -1,
$$

which gives

$$
\eta \leq \frac{2}{\lambda_{\max}(\boldsymbol{A}) + \lambda_\mathsf{P}}, \quad \lambda_i + \lambda_\mathsf{P} \geq 0.
$$

For $\mathbb{E}[\boldsymbol{C}_t \otimes \boldsymbol{C}_t]$, we have:

$$
\begin{aligned}
\mathbb{E}[\boldsymbol{C}_t \otimes \boldsymbol{C}_t] &= \mathbb{E}[(\boldsymbol{I} - \eta\boldsymbol{A}_t) \otimes (\boldsymbol{I} - \eta\boldsymbol{A}_t)] \\
&= \boldsymbol{I} - \eta\boldsymbol{A} \otimes \boldsymbol{I} - \eta\boldsymbol{I} \otimes \boldsymbol{A} + \mathbb{E}[\eta^2\boldsymbol{A}_t \otimes \boldsymbol{A}_t] \\
&= (\boldsymbol{I} - \eta\boldsymbol{A}) \otimes (\boldsymbol{I} - \eta\boldsymbol{A}) + \eta^2\mathsf{Cov}(\boldsymbol{A}_t).
\end{aligned}
$$

We observe that $\mathsf{Cov}(\boldsymbol{A}_t)$ is positive semi-definite (PSD). The spectral radius of the first term is $\max_i(1 - \eta\lambda_i - \eta\lambda_\mathsf{P})^2$. The stability similarly gives two conditions:

$$
\begin{aligned}
\lambda_i + \lambda_\mathsf{P} &\geq 0 \\
\eta &\leq \eta_U,
\end{aligned}
$$

where $\eta_U$ is some constant dependent on $\sigma, \lambda_{\max}(C)$. The second inequality for $\eta$ comes from a crucial observation: when $\eta \leq \max_{i \in [d]} \left\{ \frac{\lambda_i}{\lambda_i + \sigma^2 + \sigma} \right\}$, where $\sigma^2 := \rho(\mathsf{Cov}(A_t))$ then

$$\rho\left((I - \eta A) \otimes (I - \eta A) + \eta^2 \mathsf{Cov}(A_t)\right) \leq \max\left\{(1 - \eta(\lambda_i + \lambda_P))^2 + \eta^2 \sigma^2\right\}.$$

By the upper bound of $\eta$, one can directly verify that for any $i \in [d]$,

$$1 - \eta(\lambda_i + \lambda_P))^2 + \eta^2 \sigma^2 \leq 1,$$

Thus we conclude that when $\eta \leq \min_{i \in [d]} \left\{ \frac{2}{\lambda_{\max}(A) + \lambda_P}, \max_{i \in [d]} \left\{ \frac{\lambda_i}{\lambda_i + \sigma^2 + \sigma} \right\} \right\}$, the system is stable, thus there exists an supremum $\eta_\star$ for any $\eta \leq lr_\star$, the system is stable. Taking $\lambda_P = 0$ completes the proof of the original Corollary H.4 and Corollary H.2. And actually, we have finished the stability analysis for the system with $\lambda > 0$, which is introduced in Appendix H.1.3. $\qquad \square$

## K   EoS Detection Error for AGMs

For the proof of Theorem H.8 and Theorem H.10, we provide a unifed theorem covering both these two as shown below.

**Theorem K.1** (EoS Detection Error for RMSProp). *Given the Batch covariance matrix* $\mathsf{Cov}(\nabla^2 \mathcal{L}_{\mathcal{B}_t}(w_{t_0})) \preceq \sigma I$, *consider AGMs in the form of Equations* (15) *to* (17) *with* $a = 0$, *it holds that*

$$\frac{\eta_\star^{\lambda_P}}{\eta_\star^0} \in \left[ \frac{\mu_{\max}(\mu_{\max} + \lambda_P)}{(\mu_{\max} + \lambda_P)^2 + \sigma^2}, \frac{\mu_{\max}^2 + \sigma^2}{\mu_{\max}(\mu_{\max} + \lambda_P)} \right],$$

*where* $\mu_{\max}, \mu_{\min}$ *represent the maximum and minimal eigenvalues of* $P^{-1} \nabla^2 \mathcal{L}(w_{t_0}) + \lambda_{\mathrm{WD}} I$. *Specifically, if* $\lambda_P = -\mu_{\min}$, *then we have*

$$\frac{\eta_\star^{\lambda_P}}{\eta_\star^0} \in \left[ \frac{\kappa^2 + \kappa}{(\kappa + 1)^2 + \alpha}, \frac{\kappa^2 + \alpha^2}{\kappa^2 + \kappa} \right]$$

*where* $\alpha = \frac{\sigma^2}{|\mu_{\min}|^2}$ , $\kappa = \mu_{\max}/|\mu_{\min}|$.

This theorem is basically a replication of Theorem H.10. One can see that Theorem H.8 is a direct corollary of Theorem K.1 by setting $P = I$.

*Proof of Theorem K.1.* We first write out the update rule of RMSProp with a weight decay parameter $\lambda_{\mathrm{WD}}$:

$$\theta_{t+1} = \theta_t - \eta P^{-1}(g_t + A_t \theta_t) - \eta \lambda_{\mathrm{WD}}(\theta_t + w_{t_0}).$$

Let $\epsilon_t = \theta_t - x_*$, then we have

$$\begin{bmatrix} m_{t+1}^{(1)} \\ m_{t+1}^{(2)} \end{bmatrix} = \begin{bmatrix} C, & 0 \\ \eta S, & \mathbb{E}[C_t \otimes C_t] \end{bmatrix} \begin{bmatrix} m_t^{(1)} \\ m_t^{(2)} \end{bmatrix} + \begin{bmatrix} 0 \\ \eta^2 b \end{bmatrix},$$

where $C = \left(I - \eta \left(P^{-1} H + \lambda_P I + \lambda_{\mathrm{WD}} I\right)\right)$. Recall that our goal is to get the bound for $\frac{\eta_\star^{\lambda_P}}{\eta_\star^0}$ The assumption $\preceq \mathsf{Cov}(H_t) \preceq \sigma^2 I$ gives the upper bound and lower bound for $\eta_\star^{\lambda_P}$:

**Upper bound**: $\eta_\star^{\lambda_P} \leq \eta_U^{\lambda_P}$, where $\eta_U^{\lambda_P} := \sup\{\eta : \rho(M(\eta; \lambda_P)) < 1\}$.

**Lower bound:** $\eta_\star^{\lambda_P} \geq \eta_L^{\lambda_P}$, where $\eta_L^{\lambda_P} := \sup\left\{\eta : \|M(\eta; \lambda_P)\|^2 + \eta^2 \sigma^2 < 1\right\}$.

So we have

$$\eta_L^{\lambda_P} \leq \eta_\star^{\lambda_P} \leq \eta_U^{\lambda_P}, \quad \eta_L^0 \leq \eta_\star^0 \leq \eta_U^0.$$

And

$$\frac{\eta_L^{\lambda_P}}{\eta_U^0} \leq \frac{\eta_\star^{\lambda_P}}{\eta_\star^0} \leq \frac{\eta_U^{\lambda_P}}{\eta_L^0}.$$

We denote the eigenvalues of $\boldsymbol{P}^{-1}\boldsymbol{H} + \lambda_{\mathrm{WD}}\boldsymbol{I}$ by $\mu_1, \mu_2, \ldots, \mu_d$,

$$\eta_U^{\lambda_{\mathsf{P}}} = \frac{2}{\mu_{\max} + \lambda_{\mathsf{P}}}$$

and

$$\eta_L^{\lambda_{\mathsf{P}}} = \frac{2(\mu_{\max} + \lambda_{\mathsf{P}})}{(\mu_{\max} + \lambda_{\mathsf{P}})^2 + \sigma_2^2}$$

$$\frac{\eta_L^{\lambda_{\mathsf{P}}}}{\eta_U^0} = \frac{\mu_{\max}(\mu_{\max} + \lambda_{\mathsf{P}})}{(\mu_{\max} + \lambda_{\mathsf{P}})^2 + \sigma^2}$$

Taking $\lambda_{\mathsf{P}} = |\mu_{\min}|$ gives

$$\frac{\eta_L^{\lambda_{\mathsf{P}}}}{\eta_U^0} = \frac{\kappa^2 + \kappa}{(\kappa + 1)^2 + \alpha^2},$$

where $\kappa = \frac{\mu_{\max}}{|\mu_{\min}|}$ and $\alpha = \frac{\sigma}{|\mu_{\min}|}$.

Similarly we have

$$\frac{\eta_U^{\lambda_{\mathsf{P}}}}{\eta_L^0} = \frac{\mu_{\max}^2 + \sigma^2}{\mu_{\max}(\mu_{\max} + \lambda_{\mathsf{P}})}.$$

Taking $\lambda_{\mathsf{P}} = |\mu_{\min}|$ gives

$$\frac{\eta_U^{\lambda_{\mathsf{P}}}}{\eta_L^0} = \frac{\kappa^2 + \alpha^2}{\kappa^2 + \kappa}.$$

Thus we have $\frac{\eta_\star^{\lambda_{\mathsf{P}}}}{\eta_\star^0} = 1(1 + O(\frac{1}{\kappa}))$. □

## L   LINEAR SYSTEM DERIVATION FOR AGMS

For the local simulation of AGMs, we write out the update rule as

$$\mathbf{m}_{t+1} = a \cdot \mathbf{m}_t + (1 - a) \cdot (\boldsymbol{g}_t + \boldsymbol{A}_t \boldsymbol{\theta}_t)$$
$$\boldsymbol{\theta}_{t+1} = \boldsymbol{\theta}_t - \eta \boldsymbol{P}^{-1} \mathbf{m}_{t+1} - \eta \lambda_{\mathrm{WD}}(\boldsymbol{\theta}_t + \boldsymbol{w}_{t_0}).$$

To simplify notations, we denote $a = \beta_1$ and $b = 1 - \beta_1$. Let $\epsilon_t = \boldsymbol{\theta}_t - \boldsymbol{x}_\star$ and organizing the above equations give

$$\eta \boldsymbol{P}^{-1} \mathbf{m}_{t+1} = a\eta \boldsymbol{P}^{-1} \cdot \mathbf{m}_t + b\eta \boldsymbol{P}^{-1} \boldsymbol{A}_t \epsilon_t + b\eta \boldsymbol{P}^{-1}(\boldsymbol{g}_t + \boldsymbol{A}_t \boldsymbol{x}_\star).$$
$$\epsilon_{t+1} = (I - \eta b \boldsymbol{P}^{-1} \boldsymbol{A}_t - \eta \lambda_{\mathrm{WD}})\epsilon_t - \eta a \boldsymbol{P}^{-1} \mathbf{m}_t$$
$$- \eta(b\boldsymbol{g}_t + b\boldsymbol{A}_t \boldsymbol{x}_\star + \lambda_{\mathrm{WD}} \boldsymbol{w}_{t_0} + \lambda_{\mathrm{WD}} \boldsymbol{x}_\star).$$

Define $\boldsymbol{D}_t = \boldsymbol{P}^{-1} \boldsymbol{H}_t$ and let

$$\boldsymbol{M}_t = \begin{bmatrix} a\boldsymbol{I} & b\eta \boldsymbol{D}_t \\ -a\boldsymbol{P}^{-1} & \boldsymbol{I} - \eta b \boldsymbol{D}_t - \eta \lambda_{\mathrm{WD}} \boldsymbol{I} \end{bmatrix},$$

and

$$\boldsymbol{q}_t = \begin{bmatrix} b\eta \boldsymbol{P}^{-1}(\boldsymbol{g}_t + \boldsymbol{A}_t \boldsymbol{x}_\star) \\ -\eta(b\boldsymbol{g}_t + b\boldsymbol{A}_t \boldsymbol{x}_\star + \lambda_{\mathrm{WD}} \boldsymbol{w}_{t_0} + \lambda_{\mathrm{WD}} \boldsymbol{x}_\star). \end{bmatrix}$$

Let $\boldsymbol{z}_t = \left((\eta \boldsymbol{P}^{-1} \mathbf{m}_t)^\top, \epsilon_t^\top\right)^\top$, then we have the linear system

$$\boldsymbol{z}_{t+1} = \boldsymbol{M}_t \boldsymbol{z}_t + \boldsymbol{q}_t.$$

We denote $\boldsymbol{m}_t^{(1)} = \mathbb{E}[\boldsymbol{z}_t]$, thus we have

$$\boldsymbol{m}_{t+1}^{(1)} = \boldsymbol{M} \boldsymbol{m}_t^{(1)},$$

where

$$M = \begin{bmatrix} \beta_1 I & b\eta D \\ -aP^{-1} & I - \eta bD - \eta \lambda_{\mathrm{WD}} I \end{bmatrix}$$

We the define the second moment matrix $\Sigma_t := \mathbb{E}[z_t z_t^\top]$ and the second moment vector $m_t^{(2)} = \mathrm{vec}(\Sigma_t)$. Then we write out the second moment evolution

$$m_{t+1}^{(2)} = \mathbb{E}[M_t \otimes M_t] m_t^{(2)} + S m_t^{(1)} + b,$$

where

$$S := \mathbb{E}[q_t \otimes M_t + M_t \otimes q_t], \quad b := \mathrm{vec}(\mathbb{E}[q_t q_t^\top]).$$

So the evolution for this linear system can be written as

$$\begin{bmatrix} m_{t+1}^{(1)} \\ m_{t+1}^{(2)} \end{bmatrix} = \begin{bmatrix} M, & 0 \\ S, & \mathbb{E}[M_t \otimes M_t] \end{bmatrix} \begin{bmatrix} m_t^{(1)} \\ m_t^{(2)} \end{bmatrix} + \begin{bmatrix} 0 \\ b \end{bmatrix}.$$

We denote $\rho(M)$ the spectral radius of $M$. Since the transition matrix is block lower triangular, we have

$$\rho\left(\begin{bmatrix} M & 0 \\ S & \mathbb{E}[M_t \otimes M_t] \end{bmatrix}\right) = \max\{\rho(M), \rho(\mathbb{E}[M_t \otimes M_t])\}.$$

In fact, we know that

$$\mathbb{E}[M_t \otimes M_t] = M \otimes M + \mathrm{Cov}(M_t).$$

# M   MINIMAL EIGENVALUE ESTIMATION WITH STOCHASTIC POWER ITERATION

## M.1   NOTATION AND PROBLEM SETUPS

Before we start proving Theorem H.6, we first recall some necessary notations and clarify our problem setups.

Let $A \in \mathbb{R}^{d \times d}$ be symmetric with eigenvalues $\lambda_1(A) \geq \cdots \geq \lambda_d(A) =: \lambda_{\min}(A)$. Define the compensation threshold

$$\lambda_\star := \inf\{\lambda \geq 0 : A + \lambda I \succeq 0\} = \max\{0, -\lambda_{\min}(A)\}.$$

For each step $t$, we observe symmetric matrices

$$A_t = A + \Delta_t, \qquad \mathbb{E}[\Delta_t \mid \mathcal{F}_{t-1}] = 0, \qquad \|\Delta_t\| \leq \sigma, \quad \text{a.s.} \tag{20}$$

where filtration $\mathcal{F}_t = \sigma(\Delta_0, \ldots, \Delta_{t-1})$. Given a candidate compensation coefficient $\lambda \in [0, \Lambda]$ and step size $\eta = \left[\omega(\frac{1}{T}), \frac{1}{\sqrt{T}}\right]$, define

$$Y_t(\lambda) := I - \eta(A_t + \lambda I), \qquad M(\lambda) := \mathbb{E}Y_t(\lambda) = I - \eta(A + \lambda I),$$

and the product

$$Z_t(\lambda) := Y_{t-1}(\lambda) \cdots Y_0(\lambda), \quad Z_0(\lambda) = I.$$

With initialization $x_0 \neq 0$, the iterates satisfy $x_t(\lambda) = Z_t(\lambda) x_0$. Also, one important property implied by the range of $\eta$ is

$$\eta(\lambda_{\max}(A) + \Lambda) \leq 1 \qquad \Longrightarrow \qquad \sup_{\lambda \in [\lambda_\star, \Lambda]} \|M(\lambda)\| \leq 1. \tag{21}$$

And this property is frequently used in our following proof.

**Explosion test and $N$-round bisection.**   Fix a threshold $B > 1$. For a given $\lambda$, we run $T$ steps and declare

$$\mathrm{Explode}(\lambda) = 1 \iff \max_{0 \leq t \leq T} \|x_t(\lambda)\|_2 > B \|x_0\|_2.$$

We run $N$ rounds of bisection over $[0, \Lambda]$ using this predicate and output $\widehat{\lambda}$ as the final upper endpoint. We assume each bisection query uses an independent fresh length-$T$ trajectory $\{A_t\}_{t=0}^{T-1}$.

Now we are ready to give the following complete statement of Theorem H.6.

**Theorem M.1** (High-probability accuracy of $\hat{\lambda}$). *Given LR* $\eta = \left[\omega(\frac{1}{T}), o\left(\frac{1}{\sqrt{T}}\right)\right]$. *Run N rounds of bisection on* $[0, \Lambda]$ *with threshold* $B \geq 2$, *where each query* $\lambda$ *uses an independent length-T trajectory of samples. Given initialization* $\boldsymbol{x}_0$ *such that there exists some absolute constant* $\varepsilon > 0$,

$$\frac{|\boldsymbol{x}_0^\top \boldsymbol{u}_{\min}|}{\|\boldsymbol{x}_0\|} \geq \varepsilon > 0,$$

*where* $\boldsymbol{u}_{\min}$ *is the corresponding eigenvector of* $\boldsymbol{A}$'s *minimal eigenvalue* $\lambda_{\min}$*Given* $\lambda_{\min}(A) < 0$, *then with probability at least* $1 - \delta$,

$$\left|\hat{\lambda} - \lambda_\star\right| \leq \tilde{\mathcal{O}}\left(\frac{1}{\eta T}\right) + \tilde{\mathcal{O}}\left(\frac{\sigma\sqrt{\log d}}{\sqrt{T}}\right) + \frac{\Lambda}{2^N}, \tag{22}$$

*In particular,*

$$\left|\hat{\lambda} - \lambda_\star\right| = \tilde{\mathcal{O}}\left(\frac{1}{\eta T}\right) + \tilde{\mathcal{O}}\left(\frac{\sigma\sqrt{\log d}}{\sqrt{T}}\right) + \mathcal{O}\left(2^{-N}\right). \qquad \text{(w.h.p.)}$$

To prove the above theorem, we consider two scenarios: (1) $\lambda \geq \lambda_\star$, (2)$\lambda < \lambda_\star$. We give the following two lemmas, which imply Theorem M.1.

**Lemma M.2** (stable when $\lambda \geq \lambda_\star$). *Assume* $\eta = \left[\omega(\frac{1}{T}), o\left(\frac{1}{\sqrt{T}}\right)\right]$ *and take* $B \geq 2$. *Fix a query* $\lambda \in [\lambda_\star, \Lambda]$. *Then, with probability at least* $1 - \delta/N$,

$$\max_{0 \leq t \leq T} \|\boldsymbol{x}_t(\lambda)\|_2 \leq B\|\boldsymbol{x}_0\|_2.$$

**Lemma M.3** (Explosion when $\lambda < \lambda_\star$). *Assume* $\eta = \left[\omega(\frac{1}{T}), o\left(\frac{1}{\sqrt{T}}\right)\right]$ *and take* $B \geq 2$. *Fix one query* $\lambda = \lambda_\star - \epsilon, \epsilon > 0$ *with initialization* $\boldsymbol{x}_0$ *such that there exists some positive constant* $\varepsilon > 0$ *independent of* $T, \eta$,

$$\frac{|\boldsymbol{x}_0^\top \boldsymbol{u}_{\min}|}{\|\boldsymbol{x}_0\|} \geq \varepsilon > 0,$$

*define* $\rho := \|\boldsymbol{M}(\lambda)\|$ *and for* $\delta \in (0, 1)$ *define*

$$\Gamma_T(\delta) := C\,\eta\sigma\left(\sqrt{T\log\frac{4dTN}{\delta}} + \log\frac{4dTN}{\delta}\right). \tag{23}$$

*where* $C > 0$ *is a sufficiently large absolute constant independent of* $T, \eta$. *Fix* $\delta \in (0, 1)$. *Then, with probability at least* $1 - \delta/N$,

$$\frac{\|\boldsymbol{x}_T(\lambda)\|_2}{\|\boldsymbol{x}_0\|_2} \geq \rho^T \varepsilon\left((1 - 2\Gamma_T(\delta))\right).$$

*Consequently, if*

$$\rho^T \geq \frac{B}{\varepsilon(1 - 2\Gamma_T(\delta))},$$

*then* $\text{Explode}(\lambda) = 1$. *In particular, since* $\rho \geq 1 + \eta\epsilon$, *it suffices that*

$$(1 + \eta\epsilon)^T \geq \frac{B}{\varepsilon(1 - 2\Gamma_T(\delta))}.$$

*Equivalently, defining the (random-init) gray-zone width*

$$\epsilon_T(\delta) := \frac{1}{\eta T}\left(\log\left(\frac{B}{\varepsilon}\right) + 2\Gamma_T(\delta)\right) = \mathcal{O}\left(\frac{1}{\eta T}\right) + \tilde{\mathcal{O}}\left(\frac{\sigma\sqrt{\log d}}{\sqrt{T}}\right), \tag{24}$$

*we have: if* $\epsilon \geq \epsilon_T(\delta)$ *then* $\text{Explode}(\lambda) = 1$ *with probability at least* $1 - \delta/N$.

Now, assuming the correctness of the above Lemmas, which we will prove later, we use these two Lemmas to prove Theorem M.1.

*Proof of Theorem M.1.* By Lemma M.2, for any queried $\lambda \geq \lambda_\star$ the procedure is stable (no explosion) with probability at least $1 - \delta/N$. By Lemma M.3, for any queried $\lambda \leq \lambda_\star - \varepsilon_T(\delta)$ the procedure explodes on the same high-probability event. Therefore, on an event of probability at least $1 - \delta$ (union bound over $N$ queries), the explode/stable predicate is correct outside the interval $[\lambda_\star - \varepsilon_T(\delta), \lambda_\star + \varepsilon_T(\delta)]$, which directly implies the output $\hat{\lambda}$ lies within $\varepsilon_T(\delta)$ of $\lambda_\star$ up to the bisection resolution $\Lambda/2^N$, yielding Equation (22). $\qquad\square$

## M.3 PROOF OF LEMMA M.2

Fix $B > 1$ and define the operator-norm stopping time

$$\tau(\lambda) := \inf\{t \in \{0, 1, \ldots, T\} : \|\boldsymbol{Z}_t(\lambda)\| > B\} \wedge T. \qquad (25)$$

By definition,

$$\|\boldsymbol{Z}_s(\lambda)\| \leq B \qquad \text{for all integers } s \in \{0, 1, \ldots, \tau(\lambda) - 1\}. \qquad (26)$$

We then give an important lemma below.

**Lemma M.4** (Stopping-time matrix-Freedman concentration). *Assume* (20) *and* (21) *hold. Fix a query* $\lambda \in [0, \Lambda]$ *and define* $\tau = \tau(\lambda)$ *as in* (25). *Then for any* $\delta \in (0, 1)$, *with probability at least* $1 - \delta/N$,

$$\|\boldsymbol{Z}_t(\lambda) - \boldsymbol{M}(\lambda)^t\| \leq B\,\Gamma_T(\delta) \qquad \text{for all } t \in \{0, 1, \ldots, T\} \text{ such that } t \leq \tau(\lambda). \qquad (27)$$

*Consequently, by a union bound over the $N$ bisection queries (each using an independent trajectory), Equation* (27) *holds simultaneously for all $N$ queries with probability at least $1 - \delta$.*

To prove the above lemma, we first give two technical lemmas

**Lemma M.5** (Self-adjoint dilation). *For any (real) matrix $X \in \mathbb{R}^{d \times d}$ define its self-adjoint dilation*

$$\mathcal{D}(X) := \begin{pmatrix} 0 & X \\ X^\top & 0 \end{pmatrix} \in \mathbb{R}^{2d \times 2d}.$$

*Then $\mathcal{D}(X)$ is symmetric and*

$$\|\mathcal{D}(X)\| = \|X\|, \qquad \mathcal{D}(X)^2 = \begin{pmatrix} XX^\top & 0 \\ 0 & X^\top X \end{pmatrix}.$$

**Theorem M.6** (Matrix Freedman (Theorem 1.2 in Tropp (2011))). *Let $\{\boldsymbol{S}_k\}_{k \geq 0}$ be a self-adjoint matrix martingale of dimension $m$ with difference sequence $\boldsymbol{X}_k := \boldsymbol{S}_k - \boldsymbol{S}_{k-1}$ adapted to a filtration $\{\mathcal{F}_k\}$. Assume $\mathbb{E}[\boldsymbol{X}_k \mid \mathcal{F}_{k-1}] = 0$ and $\lambda_{\max}(\boldsymbol{X}_k) \leq R$ almost surely for all $k$. Define the predictable quadratic variation*

$$V_n := \sum_{k=1}^n \mathbb{E}[\boldsymbol{X}_k^2 \mid \mathcal{F}_{k-1}].$$

*Then for all $u \geq 0$,*

$$\mathbb{P}\Big\{\lambda_{\max}(\boldsymbol{S}_n - \boldsymbol{S}_0) \geq u \text{ and } \|V_n\| \leq v\Big\} \leq m \cdot \exp\left(-\frac{u^2}{2(v + Ru/3)}\right).$$

*In particular, there exists an absolute constant $C_0 > 0$ such that for any $\rho \in (0, 1)$, with probability at least $1 - \rho$,*

$$\|\boldsymbol{S}_n - \boldsymbol{S}_0\| \leq C_0\Big(\sqrt{v \log \frac{m}{\rho}} + R \log \frac{m}{\rho}\Big). \qquad (28)$$

*Proof of Lemma M.4.* Fix one query $\lambda \in [0, \Lambda]$ and abbreviate $\boldsymbol{Y}_t := \boldsymbol{Y}_t(\lambda)$, $M := M(\lambda)$, $\boldsymbol{Z}_t := \boldsymbol{Z}_t(\lambda)$, $\tau := \tau(\lambda)$. Let $\mathcal{F}_s := \sigma(\Delta_0, \ldots, \Delta_{s-1}) = \sigma(\boldsymbol{Y}_0, \ldots, \boldsymbol{Y}_{s-1})$ be the natural filtration.

Fix an integer $t \in \{1, \ldots, T\}$. Define, for $s = 0, 1, \ldots, t$,

$$\boldsymbol{H}_s^{(t)} := \mathbb{E}[\boldsymbol{Z}_t \mid \mathcal{F}_s]. \qquad (29)$$

Then $\{\boldsymbol{H}_s^{(t)}\}_{s=0}^t$ is a matrix martingale w.r.t. $\{\mathcal{F}_s\}$ with $H_0^{(t)} = \mathbb{E}\boldsymbol{Z}_t$ and $H_t^{(t)} = \boldsymbol{Z}_t$.

We claim that $\boldsymbol{H}_s^{(t)}$ admits the explicit expression

$$\boldsymbol{H}_s^{(t)} = \boldsymbol{M}^{t-s}\boldsymbol{Z}_s, \qquad s = 0, 1, \ldots, t. \tag{30}$$

Indeed, by independence of the future factors $\{\boldsymbol{Y}_s, \ldots, \boldsymbol{Y}_{t-1}\}$ from $\mathcal{F}_s$ and $\mathbb{E}\boldsymbol{Y}_j = M$ for each $j$,

$$\mathbb{E}[\boldsymbol{Z}_t \mid \mathcal{F}_s] = \mathbb{E}[\boldsymbol{Y}_{t-1}\cdots\boldsymbol{Y}_s]\,\boldsymbol{Z}_s = \boldsymbol{M}^{t-s}\boldsymbol{Z}_s,$$

proving (30). In particular, $\boldsymbol{H}_0^{(t)} = \boldsymbol{M}^t$ and $\boldsymbol{H}_t^{(t)} = \boldsymbol{Z}_t$.

Define the stopped martingale

$$\widetilde{\boldsymbol{H}}_s^{(t)} := \boldsymbol{H}_{s\wedge\tau\wedge t}^{(t)}, \qquad s = 0, 1, \ldots, t,$$

and its differences $\widetilde{\boldsymbol{D}}_s^{(t)} := \widetilde{\boldsymbol{H}}_s^{(t)} - \widetilde{\boldsymbol{H}}_{s-1}^{(t)}$ for $s = 1, \ldots, t$. Then $\{\widetilde{\boldsymbol{H}}_s^{(t)}\}$ is a martingale and $\sum_{s=1}^t \widetilde{\boldsymbol{D}}_s^{(t)} = \widetilde{\boldsymbol{H}}_t^{(t)} - \widetilde{\boldsymbol{H}}_0^{(t)}$.

Using (30) and $\boldsymbol{Z}_s = \boldsymbol{Y}_{s-1}\boldsymbol{Z}_{s-1}$, the (un-stopped) increment is

$$\boldsymbol{H}_s^{(t)} - \boldsymbol{H}_{s-1}^{(t)} = \boldsymbol{M}^{t-s}\boldsymbol{Z}_s - \boldsymbol{M}^{t-(s-1)}\boldsymbol{Z}_{s-1} = \boldsymbol{M}^{t-s}(\boldsymbol{Y}_{s-1} - \boldsymbol{M})\boldsymbol{Z}_{s-1}. \tag{31}$$

From (20) we have

$$\boldsymbol{Y}_{s-1} - \boldsymbol{M} = -\eta\Delta_{s-1}, \qquad \|\boldsymbol{Y}_{s-1} - \boldsymbol{M}\| \le \eta\sigma \quad \text{a.s.} \tag{32}$$

Moreover, by (21), $\|M\| \le 1$ and hence $\|\boldsymbol{M}^{t-s}\| \le 1$. Finally, by the stopping rule (25)–(26), whenever $s - 1 < \tau$ we have $\|\boldsymbol{Z}_{s-1}\| \le B$. Therefore, combining (31)–(32),

$$\|\boldsymbol{H}_s^{(t)} - \boldsymbol{H}_{s-1}^{(t)}\| \le \|\boldsymbol{M}^{t-s}\|\,\|\boldsymbol{Y}_{s-1} - \boldsymbol{M}\|\,\|\boldsymbol{Z}_{s-1}\| \le \eta\sigma B \quad \text{on the event } \{s - 1 < \tau\}.$$

On the complementary event $\{s - 1 \ge \tau\}$ the stopped increment $\widetilde{\boldsymbol{D}}_s^{(t)}$ is identically zero. Hence, we obtain the uniform almost sure bound

$$\|\widetilde{\boldsymbol{D}}_s^{(t)}\| \le R := \eta\sigma B \qquad \text{for all } s = 1, \ldots, t, \text{ a.s.} \tag{33}$$

We bound the conditional second moment. On the event $\{s - 1 < \tau\}$. We use (31):

$$(\boldsymbol{H}_s^{(t)} - \boldsymbol{H}_{s-1}^{(t)})(\boldsymbol{H}_s^{(t)} - \boldsymbol{H}_{s-1}^{(t)})^\top = \boldsymbol{M}^{t-s}(\boldsymbol{Y}_{s-1} - \boldsymbol{M})\,\boldsymbol{Z}_{s-1}\boldsymbol{Z}_{s-1}^\top\,(\boldsymbol{Y}_{s-1} - \boldsymbol{M})^\top \boldsymbol{M}^{t-s\,\top}.$$

Conditioning on $\mathcal{F}_{s-1}$ makes $\boldsymbol{Z}_{s-1}$ measurable while $(\boldsymbol{Y}_{s-1} - \boldsymbol{M}) = -\eta\Delta_{s-1}$ is independent of $\mathcal{F}_{s-1}$. Using $\boldsymbol{Z}_{s-1}\boldsymbol{Z}_{s-1}^\top \preceq \|\boldsymbol{Z}_{s-1}\|^2 I \preceq B^2 I$ on $\{s - 1 < \tau\}$ yields

$$\mathbb{E}\Big[(\boldsymbol{H}_s^{(t)} - \boldsymbol{H}_{s-1}^{(t)})(\boldsymbol{H}_s^{(t)} - \boldsymbol{H}_{s-1}^{(t)})^\top \mid \mathcal{F}_{s-1}\Big] \preceq B^2\,\boldsymbol{M}^{t-s}\,\mathbb{E}\big[(\boldsymbol{Y}_{s-1} - \boldsymbol{M})(\boldsymbol{Y}_{s-1} - \boldsymbol{M})^\top\big]\,\boldsymbol{M}^{t-s\,\top}.$$

By (32),

$$\mathbb{E}\big[(\boldsymbol{Y}_{s-1} - \boldsymbol{M})(\boldsymbol{Y}_{s-1} - \boldsymbol{M})^\top\big] = \eta^2\,\mathbb{E}[\Delta_{s-1}^2] \preceq \eta^2\sigma^2 I,$$

where the last step uses $\Delta_{s-1}^2 \preceq \|\Delta_{s-1}\|^2 I \preceq \sigma^2 I$ a.s. Using again $\|\boldsymbol{M}^{t-s}\| \le 1$, we conclude that on $\{s - 1 < \tau\}$,

$$\Big\|\mathbb{E}\Big[(\boldsymbol{H}_s^{(t)} - \boldsymbol{H}_{s-1}^{(t)})(\boldsymbol{H}_s^{(t)} - \boldsymbol{H}_{s-1}^{(t)})^\top \mid \mathcal{F}_{s-1}\Big]\Big\| \le \eta^2\sigma^2 B^2.$$

On $\{s - 1 \ge \tau\}$ the stopped increment is 0, so the same bound holds trivially. Therefore,

$$\Big\|\sum_{s=1}^t \mathbb{E}\Big[\widetilde{\boldsymbol{D}}_s^{(t)}\widetilde{\boldsymbol{D}}_s^{(t)\,\top} \mid \mathcal{F}_{s-1}\Big]\Big\| \le t\,\eta^2\sigma^2 B^2. \tag{34}$$

An identical argument gives the same bound for $\sum \mathbb{E}[\widetilde{\boldsymbol{D}}_s^{(t)\,\top}\widetilde{\boldsymbol{D}}_s^{(t)} \mid \mathcal{F}_{s-1}]$.

The process $\widetilde{\boldsymbol{H}}_s^{(t)}$ need not be self-adjoint, so we apply Theorem M.6 to the self-adjoint dilation $\mathcal{D}(\widetilde{\boldsymbol{H}}_s^{(t)})$. Let $\widetilde{S}_s := \mathcal{D}(\widetilde{\boldsymbol{H}}_s^{(t)})$ and $\widetilde{X}_s := \widetilde{S}_s - \widetilde{S}_{s-1} = \mathcal{D}(\widetilde{\boldsymbol{D}}_s^{(t)})$. By Lemma M.5, $\|\widetilde{X}_s\| = \|\widetilde{\boldsymbol{D}}_s^{(t)}\| \le R$ and the predictable quadratic variation satisfies

$$\Big\|\sum_{s=1}^t \mathbb{E}[\widetilde{X}_s^2 \mid \mathcal{F}_{s-1}]\Big\| = \max\Big\{\Big\|\sum_{s=1}^t \mathbb{E}[\widetilde{\boldsymbol{D}}_s^{(t)}\widetilde{\boldsymbol{D}}_s^{(t)\,\top} \mid \mathcal{F}_{s-1}]\Big\|, \Big\|\sum_{s=1}^t \mathbb{E}[\widetilde{\boldsymbol{D}}_s^{(t)\,\top}\widetilde{\boldsymbol{D}}_s^{(t)} \mid \mathcal{F}_{s-1}]\Big\|\Big\} \le t\,\eta^2\sigma^2 B^2.$$

Thus Theorem M.6 (in the inverted form (28)) with dimension $m = 2d$, variance proxy $v = t\eta^2\sigma^2 B^2$, and increment bound $R = \eta\sigma B$ yields: with probability at least $1 - \rho$,

$$\|\widetilde{\boldsymbol{H}}_{t\wedge\tau}^{(t)} - \widetilde{\boldsymbol{H}}_0^{(t)}\| = \|\widetilde{\boldsymbol{H}}_{t\wedge\tau}^{(t)} - \boldsymbol{H}_0^{(t)}\| \leq C_0\Big(\sqrt{t\,\eta^2\sigma^2 B^2 \log\frac{2d}{\rho}} + \eta\sigma B \log\frac{2d}{\rho}\Big)$$

$$= C_0\,\eta\sigma B\Big(\sqrt{t\log\frac{2d}{\rho}} + \log\frac{2d}{\rho}\Big). \tag{35}$$

Recall $H_0^{(t)} = M^t$. Also, if $\tau \geq t$, then $t \wedge \tau = t$ and $H_t^{(t)} = \boldsymbol{Z}_t$, hence $\widetilde{\boldsymbol{H}}_{t\wedge\tau}^{(t)} = \boldsymbol{Z}_t$. Therefore, on the event $\{\tau \geq t\}$,

$$\|\widetilde{\boldsymbol{H}}_{t\wedge\tau}^{(t)} - \boldsymbol{H}_0^{(t)}\| = \|\boldsymbol{Z}_t - \boldsymbol{M}^t\|.$$

Consequently,

$$\mathbb{P}\Big(\tau \geq t \text{ and } \|\boldsymbol{Z}_t - \boldsymbol{M}^t\| > u\Big) \leq \mathbb{P}\Big(\|\widetilde{\boldsymbol{H}}_{t\wedge\tau}^{(t)} - \boldsymbol{H}_0^{(t)}\| > u\Big).$$

Combining this with (35) and choosing $\rho = \delta/(TN)$ gives that with probability at least $1 - \delta/(TN)$,

$$t \leq \tau \quad \Longrightarrow \quad \|\boldsymbol{Z}_t - \boldsymbol{M}^t\| \leq C_0\,\eta\sigma B\Big(\sqrt{t\log\frac{4dTN}{\delta}} + \log\frac{4dTN}{\delta}\Big).$$

Since $t \leq T$ implies $\sqrt{t} \leq \sqrt{T}$, we further obtain the uniform implication

$$t \leq \tau \quad \Longrightarrow \quad \|\boldsymbol{Z}_t - \boldsymbol{M}^t\| \leq C_0\,\eta\sigma B\Big(\sqrt{T\log\frac{4dTN}{\delta}} + \log\frac{4dTN}{\delta}\Big). \tag{36}$$

Let $\mathcal{E}_t$ be the event that (36) holds for this particular $t$. We have $\mathbb{P}(\mathcal{E}_t) \geq 1 - \delta/(TN)$. By a union bound over $t = 1, \ldots, T$, with probability at least $1 - \delta/N$, (36) holds simultaneously for all $t \in \{1, \ldots, T\}$. This is exactly (27) after enlarging $C$ in the definition of $\Gamma_T(\delta)$ in (23). (Also $t = 0$ is trivial because $\boldsymbol{Z}_0 = \boldsymbol{M}^0 = \boldsymbol{I}$.)

Also, for each query, we apply the above argument with failure probability $\delta/N$ (implemented by taking $\rho = \delta/(TN)$ inside the proof). A final union bound over the $N$ queries yields a simultaneous success event of probability at least $1 - \delta$. $\qquad\square$

For fixed $(d, N, \delta, \sigma)$,

$$\Gamma_T(\delta) = C\eta\sigma\Big(\sqrt{T\log(4dTN/\delta)} + \log(4dTN/\delta)\Big) \to 0 \quad \text{whenever } \eta = o(1/\sqrt{T}).$$

Hence, on the non-explosion segment ($t \leq \tau(\lambda)$), the deviation $\|\boldsymbol{Z}_t(\lambda) - \boldsymbol{M}(\lambda)^t\|$ vanishes in high probability as $T \to \infty$. So that we can assume that $\Gamma_T(\delta) \leq 1/4$ for the rest of our proof, which is implied naturally by the big-O notation.

*Proof of Lemma M.2.* On the event of Lemma M.4, apply Equation (27) at $t = \tau(\lambda)$ (which satisfies $t \leq \tau(\lambda)$ trivially):

$$\|\boldsymbol{Z}_\tau - \boldsymbol{M}(\lambda)^\tau\| \leq B\Gamma_T(\delta).$$

Because $\lambda \geq \lambda_\star$ and Equation (21) holds, $\|M(\lambda)^\tau\| \leq 1$. Thus

$$\|\boldsymbol{Z}_\tau\| \leq \|\boldsymbol{M}(\lambda)^\tau\| + \|\boldsymbol{Z}_\tau - \boldsymbol{M}(\lambda)^\tau\| \leq 1 + B\Gamma_T(\delta) \leq 1 + \frac{B}{4} \leq B,$$

where we used $\Gamma_T(\delta) \leq 1/4$ and $B \geq 2$. This contradicts the definition of $\tau(\lambda)$ unless $\tau(\lambda) = T$. Finally, $\|x_t\| \leq \|\boldsymbol{Z}_t\|\,\|\boldsymbol{x}_0\| \leq B\|\boldsymbol{x}_0\|$ for all $t \leq T$. $\qquad\square$

## M.4 PROOF OF LEMMA M.3

Define the normalized matrices
$$\widetilde{\boldsymbol{Y}}_t := \rho^{-1}\boldsymbol{Y}_t, \qquad \widetilde{\boldsymbol{M}} := \rho^{-1}\boldsymbol{M}, \qquad \widetilde{\boldsymbol{Z}}_t := \widetilde{\boldsymbol{Y}}_{t-1}\cdots\widetilde{\boldsymbol{Y}}_0 = \rho^{-t}\boldsymbol{Z}_t.$$

Then $\|\widetilde{\boldsymbol{M}}\| = 1$. Moreover, by the condition $\boldsymbol{A}_t = \boldsymbol{A} + \Delta_t$ with $\|\Delta_t\| \leq \sigma$,
$$\|\widetilde{\boldsymbol{Y}}_t - \widetilde{\boldsymbol{M}}\| = \rho^{-1}\|\boldsymbol{Y}_t - \boldsymbol{M}\| = \rho^{-1}\eta\|\Delta_t\| \leq \frac{\eta\sigma}{\rho} \qquad \text{a.s.}$$

Introduce the (analysis) stopping time
$$\widetilde{\tau} := \inf\{t \in \{0, 1, \ldots, T\} : \|\widetilde{\boldsymbol{Z}}_t\| > 2\} \wedge T.$$

Now apply the same stopping-time matrix-Freedman argument as in Lemma M.4 to the normalized sequence $\{\widetilde{\boldsymbol{Y}}_t\}$ (whose mean operator norm is $\leq 1$) with threshold 2. This yields the existence of an event $\mathcal{E}$ with $\mathbb{P}(\mathcal{E}) \geq 1 - \delta/N$ such that

$$\|\widetilde{\boldsymbol{Z}}_t - \widetilde{\boldsymbol{M}}^t\| \leq 2 \cdot \frac{\Gamma_T(\delta)}{\rho} \qquad \text{for all } t \leq \widetilde{\tau}. \tag{37}$$

Given $\Gamma_T(\delta) \leq \frac{1}{4}$, and since $\rho \geq 1$ we have $2\Gamma_T(\delta)/\rho \leq \frac{1}{2}$. Thus on $\mathcal{E}$,

$$\|\widetilde{\boldsymbol{Z}}_{\widetilde{\tau}}\| \leq \|\widetilde{\boldsymbol{M}}^{\widetilde{\tau}}\| + \|\widetilde{\boldsymbol{Z}}_{\widetilde{\tau}} - \widetilde{\boldsymbol{M}}^{\widetilde{\tau}}\| \leq 1 + \frac{1}{2} < 2,$$

which contradicts the definition of $\widetilde{\tau}$ unless $\widetilde{\tau} = T$. Hence on $\mathcal{E}$, (37) holds at $t = T$, and

$$\|\widetilde{\boldsymbol{Z}}_t\| \geq \|\widetilde{\boldsymbol{M}}^T\| - \|\widetilde{\boldsymbol{Z}}_t - \widetilde{\boldsymbol{M}}^T\| \geq 1 - \frac{2\Gamma_T(\delta)}{\rho} \geq 1 - 2\Gamma_T(\delta).$$

Rescaling back gives
$$\|\boldsymbol{Z}_t\| = \rho^T\|\widetilde{\boldsymbol{Z}}_t\| \geq \rho^T(1 - 2\Gamma_T(\delta)). \tag{38}$$

Condition on $\boldsymbol{Z}_t$. Let $u$ be a unit top right singular vector of $\boldsymbol{Z}_t$, so that $\|\boldsymbol{Z}_t u\|_2 = \|\boldsymbol{Z}_t\|$. Let $s := x_0/\|\boldsymbol{x}_0\|_2$. By $x_0 \sim \mathcal{N}(0, I_d)$, the direction $s$ is uniform on $\mathbb{S}^{d-1}$ and independent of $\boldsymbol{Z}_t$ (and hence independent of $u$). Then

$$\frac{\|\boldsymbol{x}_T\|_2}{\|\boldsymbol{x}_0\|_2} = \|\boldsymbol{Z}_t s\|_2 \geq \|\boldsymbol{Z}_t\|\,|\boldsymbol{u}^\top s| \geq \rho^T \varepsilon\,((1 - 2\Gamma_T(\delta))$$

The above inequality completes the proof. $\qquad\square$

**Extension with Gaussain Intialization.** Further we can extend the theorem with initialization $\boldsymbol{x}_0 \sim \mathcal{N}(0, I_d)$. We use a standard spherical-cap bound: for $s \sim \text{Unif}(\mathbb{S}^{d-1})$ and any fixed unit $u$,

$$\mathbb{P}\big(|\boldsymbol{u}^\top s| \leq t\big) \leq 2t\sqrt{d}, \qquad \forall\, t \in (0, 1).$$

Taking $t = \delta_0/(2\sqrt{d})$ yields
$$\mathbb{P}\left(|\boldsymbol{u}^\top s| \geq \frac{\delta_0}{2\sqrt{d}}\right) \geq 1 - \delta_0.$$

Combine this with (38): on the intersection of the event $\mathcal{E}$ and the above anti-concentration event (probability at least $1 - \delta/N - \delta_0$),

$$\frac{\|\boldsymbol{x}_T\|_2}{\|\boldsymbol{x}_0\|_2} \geq \rho^T(1 - 2\Gamma_T(\delta)) \cdot \frac{\delta_0}{2\sqrt{d}}.$$

Therefore, if $\rho^T \geq 4B\sqrt{d}/\delta_0$, then $\|\boldsymbol{x}_T\|_2 > B\|\boldsymbol{x}_0\|_2$, which implies $\text{Explode}(\lambda) = 1$ (since the test checks $\max_{0 \leq t \leq T}\|x_t\|_2$). The same argument derives a whp bound for Gaussian initialization. $\quad\square$

# N ALGORITHMS

---

**Algorithm 1** Minimal Eigenvalue Approximation via Stochastic Power Iteration + Bisection

---

**Require:** Stochastic matrix access via i.i.d. samples $\boldsymbol{A}_t$, step size $\eta$, inner horizon $T$, bisection rounds $N$, explosion threshold $B > 1$, initial search range $[0, \Lambda]$ (with $\Lambda$ large enough), and initialization $\boldsymbol{x}_0 \sim \mathcal{N}(0, I)$.

**Ensure:** Estimated shift $\hat{\lambda}$ and minimal eigenvalue estimate $\widehat{\lambda}_{\min} = -\hat{\lambda}$.

1: **Subroutine** EXPLODETEST($\lambda$):
2:     Sample $\boldsymbol{x}_0 \sim \mathcal{N}(0, I)$ and normalize; set $\boldsymbol{x} \leftarrow \boldsymbol{x}_0$.
3: **for** $t = 0, 1, \ldots, T-1$ **do**
4:     Draw an independent sample $\boldsymbol{A}_t$.
5:     $\boldsymbol{x} \leftarrow \boldsymbol{x} - \eta(\boldsymbol{A}_t + \lambda I)\boldsymbol{x}$.
6:     **if** $\|\boldsymbol{x}\|_2 > B$ **then**
7:         **return** false {explode}
8:     **end if**
9: **end for**
10: **return** true {stable}

11: **Main procedure:**
12: $\lambda_{\mathrm{lo}} \leftarrow 0, \quad \lambda_{\mathrm{hi}} \leftarrow \Lambda$.
13: **while** EXPLODETEST($\lambda_{\mathrm{hi}}$) = false **do**
14:     $\lambda_{\mathrm{hi}} \leftarrow 2\lambda_{\mathrm{hi}}$ {find a stable upper bound}
15: **end while**
16: **for** $k = 1, 2, \ldots, N$ **do**
17:     $\lambda_{\mathrm{mid}} \leftarrow (\lambda_{\mathrm{lo}} + \lambda_{\mathrm{hi}})/2$.
18:     **if** EXPLODETEST($\lambda_{\mathrm{mid}}$) = true **then**
19:         $\lambda_{\mathrm{hi}} \leftarrow \lambda_{\mathrm{mid}}$ {stable}
20:     **else**
21:         $\lambda_{\mathrm{lo}} \leftarrow \lambda_{\mathrm{mid}}$ {explode}
22:     **end if**
23: **end for**
24: $\hat{\lambda} \leftarrow \lambda_{\mathrm{hi}}, \quad \widehat{\lambda}_{\min} \leftarrow -\hat{\lambda}$.
25: **return** $\hat{\lambda}, \widehat{\lambda}_{\min}$.

---

---

**Algorithm 2** EoC Test for SGD

---

**Require:** Checkpoint $\boldsymbol{w}_{t_0}$; weight decay $\lambda_{\mathrm{WD}} \geq 0$; tolerance $\nu_{\mathrm{tol}} > 0$; Algorithm 1 parameters $(\eta_{\mathrm{PI}}, T_{\mathrm{PI}}, N_{\mathrm{PI}}, B_{\mathrm{PI}}, \Lambda_{\mathrm{PI}})$; Lanczos iterations $T_{\mathrm{LZ}}$.

**Ensure:** EoC $\in \{\texttt{true}, \texttt{false}\}$; estimates $\widehat{\lambda}_{\min}$, $\widehat{\lambda}_{\max}$, and $\widehat{\nu}$.

1: Define symmetric stochastic samples $\boldsymbol{A}_t := \nabla^2 \mathcal{L}_{\mathcal{B}_t}(\boldsymbol{w}_{t_0}) + \lambda_{\mathrm{WD}}\boldsymbol{I}$.
2: $(\widehat{\lambda}_{\mathrm{shift}}, \widehat{\lambda}_{\min}) \leftarrow$ Algorithm 1$(\boldsymbol{A}_t; \eta_{\mathrm{PI}}, T_{\mathrm{PI}}, N_{\mathrm{PI}}, B_{\mathrm{PI}}, \Lambda_{\mathrm{PI}})$.
3: $\widehat{\lambda}_{\max} \leftarrow \textsc{LanczosMaxEig}(\boldsymbol{A}_t; T_{\mathrm{LZ}})$.
4: $\widehat{\nu} \leftarrow \dfrac{\min\{0, -\widehat{\lambda}_{\min}\}}{\widehat{\lambda}_{\max}}$.
5: EoC $\leftarrow \left(\widehat{\lambda}_{\min} < 0\right) \wedge \left(\widehat{\nu} < \nu_{\mathrm{tol}}\right)$.
6: **return** EoC, $\widehat{\lambda}_{\min}$, $\widehat{\lambda}_{\max}$, $\widehat{\nu}$.

---

---

**Algorithm 3** EoS Test for SGD

---

**Require:** Checkpoint $\boldsymbol{w}_{t_0}$; training LR $\eta$; horizon $T$; explosion threshold $M$; seed $s$; increasing scaling factors $\{\alpha_k\}_{k=1}^K$; EoS tolerance $\epsilon_{\mathrm{eos}} > 0$. Also require Algorithm 1 parameters $(\eta_{\mathrm{PI}}, T_{\mathrm{PI}}, N_{\mathrm{PI}}, B_{\mathrm{PI}}, \Lambda_{\mathrm{PI}})$.

**Ensure:** Estimated critical LR $\widehat{\eta}^{\star}_{\lambda_{\mathsf{P}}}(\boldsymbol{w}_{t_0})$, ratio $r(\boldsymbol{w}_{t_0})$, and EOS flag.

1: **(Step 1: estimate convexity compensation.)** Define stochastic samples $\boldsymbol{A}_t := \nabla^2 \mathcal{L}_{\mathcal{B}_t}(\boldsymbol{w}_{t_0})$ via Hessian–vector products. Run Algorithm 1 on $\boldsymbol{A}_t$ to obtain $\widehat{\lambda}_{\mathrm{shift}}$. Set $\lambda_{\mathsf{P}} \leftarrow \widehat{\lambda}_{\mathrm{shift}}$.
2: **Subroutine** $\textsc{ExplodeTest}(\alpha)$:
3:      Set RNG seed to $s$; $\boldsymbol{\theta} \leftarrow 0$; $R \leftarrow 0$.
4: **for** $t = 0, 1, \ldots, T - 1$ **do**
5:      Sample mini-batch $\mathcal{B}_t$.
6:      $\boldsymbol{g} \leftarrow \nabla \mathcal{L}_{\mathcal{B}_t}(\boldsymbol{w}_{t_0}) + \nabla^2 \mathcal{L}_{\mathcal{B}_t}(\boldsymbol{w}_{t_0})\,\boldsymbol{\theta} + \lambda_{\mathsf{P}}\,\boldsymbol{\theta}$.
7:      $\boldsymbol{\theta} \leftarrow \boldsymbol{\theta} - \alpha\eta\,\boldsymbol{g}$.
8:      $R \leftarrow \max\{R, \|\boldsymbol{\theta}\|_2\}$.
9:      **if** $R \geq M$ **then**
10:         **return** $\texttt{true}$ {explode}
11:      **end if**
12: **end for**
13:      **return** $\texttt{false}$ {stable}
14: **(Step 2: search over $\{\alpha_k\}$.)** Find $k^{\star} := \min\{k : \textsc{ExplodeTest}(\alpha_k) = \texttt{true}\}$.
15: **if** no such $k^{\star}$ exists **then**
16:      $\widehat{\eta}^{\star}_{\lambda_{\mathsf{P}}} \leftarrow > \alpha_K\eta$;    $r \leftarrow 1/\alpha_K$;    EoS $\leftarrow \texttt{false}$.
17:      **return** $\widehat{\eta}^{\star}_{\lambda_{\mathsf{P}}}, r, \mathrm{EoS}$.
18: **end if**
19: Set $\alpha_{\mathrm{lo}} \leftarrow (\alpha_{k^{\star}-1}$ if $k^{\star} > 1$ else $0)$ and $\alpha_{\mathrm{hi}} \leftarrow \alpha_{k^{\star}}$.
20: $\widehat{\eta}^{\star}_{\lambda_{\mathsf{P}}}(\boldsymbol{w}_{t_0}) \leftarrow \alpha_{\mathrm{hi}}\eta$.
21: $r(\boldsymbol{w}_{t_0}) \leftarrow 1/\alpha_{\mathrm{hi}}$.
22: EoS $\leftarrow \left(r(\boldsymbol{w}_{t_0}) \in [1 - \epsilon_{\mathrm{eos}}, 1 + \epsilon_{\mathrm{eos}}]\right)$.
23: **return** $\widehat{\eta}^{\star}_{\lambda_{\mathsf{P}}}, r, \mathrm{EoS}$.

---

---

**Algorithm 4** EoC Test for AGMs

---

**Require:** Checkpoint $\boldsymbol{w}_{t_0}$; frozen preconditioner $\boldsymbol{P}_{t_0} \succ 0$; weight decay $\lambda_{\mathrm{WD}} \geq 0$; tolerance $\nu_{\mathrm{tol}} > 0$; Algorithm 1 parameters $(\eta_{\mathrm{PI}}, T_{\mathrm{PI}}, N_{\mathrm{PI}}, B_{\mathrm{PI}}, \Lambda_{\mathrm{PI}})$; Lanczos iterations $T_{\mathrm{LZ}}$.
**Ensure:** $\mathrm{EoC} \in \{\texttt{true}, \texttt{false}\}$; estimates $\widehat{\mu}_{\min}, \widehat{\mu}_{\max}$, and $\widehat{\nu}$.

1: Define symmetric stochastic samples $\widetilde{\boldsymbol{A}}_t := \boldsymbol{P}_{t_0}^{-1/2}\big(\nabla^2 \mathcal{L}_{\mathcal{B}_t}(\boldsymbol{w}_{t_0})\big)\boldsymbol{P}_{t_0}^{-1/2} + \lambda_{\mathrm{WD}}\boldsymbol{I}$.
2: $\big(\widehat{\lambda}_{\mathrm{shift}}, \widehat{\mu}_{\min}\big) \leftarrow$ Algorithm 1$(\widetilde{\boldsymbol{A}}_t; \eta_{\mathrm{PI}}, T_{\mathrm{PI}}, N_{\mathrm{PI}}, B_{\mathrm{PI}}, \Lambda_{\mathrm{PI}})$.
3: $\widehat{\mu}_{\max} \leftarrow \textsc{LanczosMaxEig}(\widetilde{\boldsymbol{A}}_t; T_{\mathrm{LZ}})$.
4: $\widehat{\nu} \leftarrow \dfrac{\min\{0, -\widehat{\mu}_{\min}\}}{\widehat{\mu}_{\max}}$.
5: $\mathrm{EoC} \leftarrow \big(\widehat{\mu}_{\min} < 0\big) \wedge \big(\widehat{\nu} < \nu_{\mathrm{tol}}\big)$.
6: **return** $\mathrm{EoC}, \widehat{\mu}_{\min}, \widehat{\mu}_{\max}, \widehat{\nu}$.

---

---

**Algorithm 5** EoS Test for AGMs

---

**Require:** Checkpoint $\boldsymbol{w}_{t_0}$; momentum state $\boldsymbol{m}_{t_0}$; frozen preconditioner $\boldsymbol{P}_{t_0} \succ 0$; weight decay $\lambda_{\mathrm{WD}} \geq 0$; training LR $\eta$; momentum parameters $\beta_1 \in [0, 1)$; horizon $T$; explosion threshold $M$; seed $s$; increasing scaling factors $\{\alpha_k\}_{k=1}^{K}$; EoS tolerance $\epsilon_{\mathrm{eos}} > 0$. Also require Algorithm 1 parameters $(\eta_{\mathrm{PI}}, T_{\mathrm{PI}}, N_{\mathrm{PI}}, B_{\mathrm{PI}}, \Lambda_{\mathrm{PI}})$.
**Ensure:** Estimated critical LR $\widehat{\eta}_{\lambda_{\mathsf{P}}}^{\star}(\boldsymbol{w}_{t_0})$, ratio $r(\boldsymbol{w}_{t_0})$, and EoS flag.

1: **(Step 1: estimate convexity compensation.)** Define symmetric stochastic samples $\widetilde{\boldsymbol{A}}_t := \boldsymbol{P}_{t_0}^{-1/2}\big(\nabla^2 \mathcal{L}_{\mathcal{B}_t}(\boldsymbol{w}_{t_0})\big)\boldsymbol{P}_{t_0}^{-1/2} + \lambda_{\mathrm{WD}}\boldsymbol{I}$. Run Algorithm 1 on $\widetilde{\boldsymbol{A}}_t$ to obtain $\widehat{\lambda}_{\mathrm{shift}}$. Set $\lambda_{\mathsf{P}} \leftarrow \widehat{\lambda}_{\mathrm{shift}}$.
2: **Subroutine** $\textsc{ExplodeTest}(\alpha)$:
3:      Set RNG seed to $s$; $\boldsymbol{\theta} \leftarrow 0$; $\boldsymbol{m} \leftarrow \boldsymbol{m}_{t_0}$; $R \leftarrow 0$.
4: **for** $t = 0, 1, \ldots, T - 1$ **do**
5:      Sample mini-batch $\mathcal{B}_t$.
6:      $\boldsymbol{g} \leftarrow \nabla \mathcal{L}_{\mathcal{B}_t}(\boldsymbol{w}_{t_0}) + \nabla^2 \mathcal{L}_{\mathcal{B}_t}(\boldsymbol{w}_{t_0})\,\boldsymbol{\theta} + \lambda_{\mathsf{P}}\,\boldsymbol{P}_{t_0}\boldsymbol{\theta}$.
7:      $\boldsymbol{m} \leftarrow \beta_1\,\boldsymbol{m} + (1 - \beta_1)\,\boldsymbol{g}$.
8:      $\boldsymbol{\theta} \leftarrow \boldsymbol{\theta} - \alpha\eta\,\boldsymbol{P}_{t_0}^{-1}\boldsymbol{m} - \alpha\eta\,\lambda_{\mathrm{WD}}(\boldsymbol{\theta} + \boldsymbol{w}_{t_0})$.
9:      $R \leftarrow \max\{R, \|\boldsymbol{\theta}\|_{\boldsymbol{P}_{t_0}}\}$.
10:    **if** $R \geq M$ **then**
11:        **return** $\texttt{true}$ {explode}
12:    **end if**
13: **end for**
14:      **return** $\texttt{false}$ {stable}
15: **(Step 2: search over $\{\alpha_k\}$.)** Find $k^{\star} := \min\{k : \textsc{ExplodeTest}(\alpha_k) = \texttt{true}\}$.
16: **if** no such $k^{\star}$ exists **then**
17:    $\widehat{\eta}_{\lambda_{\mathsf{P}}}^{\star} \leftarrow > \alpha_K \eta$;    $r \leftarrow 1/\alpha_K$;    $\mathrm{EoS} \leftarrow \texttt{false}$.
18:    **return** $\widehat{\eta}_{\lambda_{\mathsf{P}}}^{\star}, r, \mathrm{EoS}$.
19: **end if**
20: Set $\alpha_{\mathrm{lo}} \leftarrow (\alpha_{k^{\star}-1}$ if $k^{\star} > 1$ else $0)$ and $\alpha_{\mathrm{hi}} \leftarrow \alpha_{k^{\star}}$.
21: $\widehat{\eta}_{\lambda_{\mathsf{P}}}^{\star}(\boldsymbol{w}_{t_0}) \leftarrow \alpha_{\mathrm{hi}}\eta$.
22: $r(\boldsymbol{w}_{t_0}) \leftarrow 1/\alpha_{\mathrm{hi}}$.
23: $\mathrm{EoS} \leftarrow \big(r(\boldsymbol{w}_{t_0}) \in [1 - \epsilon_{\mathrm{eos}}, 1 + \epsilon_{\mathrm{eos}}]\big)$.
24: **return** $\widehat{\eta}_{\lambda_{\mathsf{P}}}^{\star}, r, \mathrm{EoS}$.

---

