# OpenReview forum: "Does LLM Pre-Training Typically Occur at the Edge of Stability?"
_ICLR.cc/2026/Workshop/Sci4DL — Sci4DL 2026_

### Official Review · Reviewer_ZGqL · 2026-02-18

**Fit:** 3
**Significance:** 2
**Confidence:** 3

**Summary:**

This work investigates the training dynamics of neural networks for language model pretraining using SGD and AdamW. Empirically, it reports two recurring phenomena:

(1) Edge of Convexity (EoC) is consistently observed: the minimum eigenvalue of the (full-batch) Hessian at training checkpoints remains negative, but its magnitude is small relative to the maximum eigenvalue (typically ≤ 10%), indicating persistent but weak negative curvature.

(2) Edge of Stability (EoS) is prevalent but not universal: after compensating for negative curvature and restricting to the nonnegative-curvature subspace, local quadratic simulations with stochastic (mini-batch) Hessians exhibit a sharp stability–instability transition as the learning rate increases, although this effect weakens under smaller learning rates or batch sizes.

To measure these phenomena at scale, the authors introduce a stochastic power-iteration-based method to estimate the minimum eigenvalue and use Lanczos with Hessian–vector products to estimate the maximum eigenvalue, enabling curvature diagnostics without costly eigendecomposition.

**Strengths:**

This paper studies optimization dynamics in LLM pretraining, an important and timely topic for the workshop. A key strength is the introduction of practical diagnostics for detecting Edge of Convexity (EoC) and Edge of Stability (EoS) that are applicable at modern scales (up to 1.7B parameters), whereas much of the prior EoS literature has focused on significantly smaller models. The work provides empirical evidence that weak but persistent negative curvature is present during training and systematically investigates when EoS emerges under stochastic and adaptive optimizers (SGD, AdamW).

**Suggestions:**

The paper is good enough in its current form for this workshop (under the 4-page limit). For submission to future venues, I would like to offer a few suggestions and questions.

1. The paper states that experiments are conducted for both SGD and AdamW, but I was only able to identify AdamW results. Moreover, according to the appendix, the maximum and minimum eigenvalues of the preconditioned Hessian are estimated for AdamW, whereas the main figures appear to show eigenvalues of the (unpreconditioned) Hessian. For clarity, it would be helpful to explicitly state for each figure (a) which optimizer is used and (b) whether the reported eigenvalues correspond to the Hessian or the preconditioned Hessian.

2. According to Figure 11 in the appendix, there are several interesting observations that may be worth highlighting in the paper. First, the eigenvector similarity corresponding to the top eigenvalue is close to one; it would be helpful to clarify how this similarity is computed. Second, during the constant LR phase near EoS, the maximum eigenvalue appears relatively stable, while during the LR decay phase it increases (i.e., progressive sharpening). It would be valuable to investigate this behavior with additional controlled experiments and discuss whether it is consistently observed across settings.

3. When changing model scale, it is common practice to scale the total number of training tokens accordingly (e.g., following Chinchilla-style scaling laws). In this work, larger models appear to be trained with fewer tokens, which is not standard and may confound comparisons across scales. I suggest adopting a more standard scaling protocol or providing justification for the chosen setup.

4. The paper currently lacks sufficient discussion of the uncertainty in the eigenvalue estimates. Providing confidence intervals, variance across runs, or sensitivity analyses would help establish the reliability of the reported curvature measurements.

---

### Official Review · Reviewer_kUuN · 2026-02-21

**Fit:** 3
**Significance:** 3
**Confidence:** 3

**Summary:**

This paper studies how modern LLM training (which involves stochastic optimization) relates to the edge of stability (EOS) phenomenon (which has mostly been studied in the context of full-batch training).  They find that LLM training often occurs "near the edge" (my words), in the sense that increasing the learning rate by a factor less than 2 is sufficient to trigger divergence on the local quadratic Taylor approximation.  The exceptions are (a) if the batch size is small enough, or (b) after a learning rate decay.  They also define a related notion, which they term "edge of convexity," related to diverging along the negative Hessian eigenvalue directions.

**Strengths:**

The paper is the first to investigate EoS-like phenomena in the context of LLM training, and offers a good perspective on EoS in stochastic optimization in general.

**Suggestions:**

* In Figure 2 and related figures, since you are presumably running Adam, it would make more sense to show the preconditioned sharpness  (https://arxiv.org/abs/2207.14484) rather than the top eigenvalue of the 'raw' Hessian.
* You say that: "NN training sits on an edge where either divergence mechanism is just at the threshold of breaking the local quadratic approximation."  I am not sure if this is accurate regarding the EoC mechanism. In what sense are we "at the edge" of diverging along the negative eigenvalue directions?  You always diverge, at any learning rate, along these directions.

---

### Official Review · Reviewer_iYqm · 2026-02-26

**Fit:** 2
**Significance:** 2
**Confidence:** 2

**Summary:**

The paper analyzes the EoS phenomena during LLM pretraining and identifies negative eigenvalues throughout training. Furthermore, they show that EoS reliably occurs during LLM pre-training for batch sizes above ~!1/3 critical batch size. The authors also propose algorithms to compute negative eigenvalues in practice.

**Strengths:**

* Identification of negative Hessian eigenvalues during LLM training

**Suggestions:**

* The results can be discussed better
* The significance of the negative eigenvalues is not very clear from the main text
* It would be interesting to see if the existence of negative eigenvalues is an initialization artifact.

---

### Meta-Review · Area_Chair_t2hc · 2026-03-01

**Recommendation:** Accept

**Metareview:**

Very positive reviews. I recommend accept.

---

### Decision · Program_Chairs · 2026-03-02

Accept